# Efficient and Accurate Explanation Estimation with Distribution Compression

**Hubert Baniecki**
University of Warsaw
`h.baniecki@uw.edu.pl`

**Giuseppe Casalicchio**
LMU Munich
Munich Center for Machine Learning

**Bernd Bischl**
LMU Munich
Munich Center for Machine Learning

**Przemyslaw Biecek**
University of Warsaw
Warsaw University of Technology

## Abstract

We discover a theoretical connection between explanation estimation and distribution compression that significantly improves the approximation of feature attributions, importance, and effects. While the exact computation of various machine learning explanations requires numerous model inferences and becomes impractical, the computational cost of approximation increases with an ever-increasing size of data and model parameters. We show that the standard i.i.d. sampling used in a broad spectrum of algorithms for post-hoc explanation leads to an approximation error worthy of improvement. To this end, we introduce *compress then explain* (CTE), a new paradigm of sample-efficient explainability. It relies on distribution compression through kernel thinning to obtain a data sample that best approximates its marginal distribution. CTE significantly improves the accuracy and stability of explanation estimation with negligible computational overhead. It often achieves an on-par explanation approximation error 2–3× faster by using fewer samples, i.e. requiring 2–3× fewer model evaluations. CTE is a simple, yet powerful, plug-in for any explanation method that now relies on i.i.d. sampling.

## 1 Introduction

Computationally efficient estimation of post-hoc explanations is at the forefront of current research on explainable machine learning (Strumbelj & Kononenko, 2010; Slack et al., 2021; Jethani et al., 2022; Chen et al., 2023; Donnelly et al., 2023; Muschalik et al., 2024). The majority of the work focuses on improving efficiency with respect to the dimension of features (Covert et al., 2020; Jethani et al., 2022; Chen et al., 2023; Fumagalli et al., 2023), specific model classes like neural networks (Erion et al., 2021; Chen et al., 2024) and decision trees (Muschalik et al., 2024), or approximating the conditional feature distribution (Chen et al., 2018; Aas et al., 2021; Olsen et al., 2022; 2024).

However, in many practical settings, a marginal feature distribution is used instead to estimate explanations, and i.i.d. samples from the data typically form the so-called *background data* samples, also known as *reference points* or *baselines*, which plays a crucial role in the estimation process (Lundberg & Lee, 2017; Scholbeck et al., 2020; Erion et al., 2021; Ghalebikesabi et al., 2021; Lundstrom et al., 2022). For example, Covert et al. (2020) mention "*[O]ur sampling approximation for SAGE was run using draws from the marginal distribution. We used a fixed set of 512 background samples [...]*" and we provide more such references in Appendix A to motivate our research question: *Can we reliably improve on standard i.i.d. sampling in explanation estimation?*

We make a connection to research on statistical theory, where kernel thinning (KT, Dwivedi & Mackey, 2021; 2022) was introduced to compress a distribution more effectively than with i.i.d. sampling. KT has an efficient implementation in the COMPRESS++ algorithm (Shetty et al., 2022) and was applied to improve statistical kernel testing (Domingo-Enrich et al., 2023). Building on this line of work, this paper aims to thoroughly quantify the error introduced by the current *sample then explain* paradigm in feature marginalization, which is involved in the estimation of both local and global removal-based explanations (Covert et al., 2021). We propose an efficient way to reduce this approximation error based on distribution compression (Figure 1).

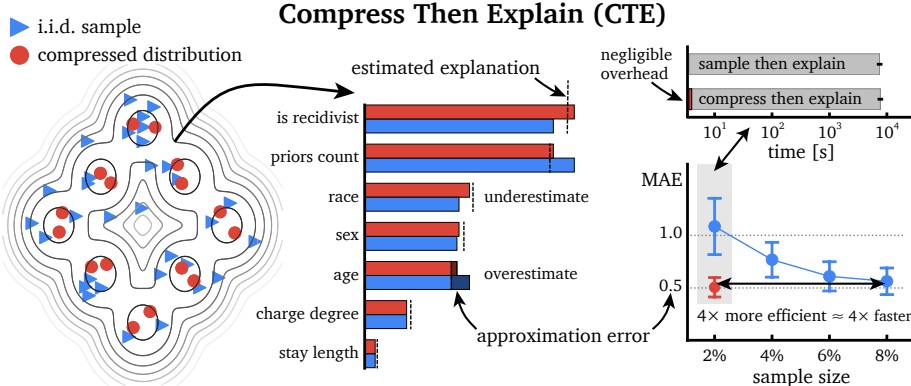

**Figure 1: Garbage sample in, garbage explanation out.** *Sample then explain* is a conventional approach to decrease the computational cost of explanation estimation. Although fast, sampling is inefficient and prone to error, which may even lead to changes in feature importance rankings. We propose *compress then explain* (CTE), a new paradigm for accurate, yet efficient, estimation of explanations based on a marginal distribution that is compressed, e.g. with kernel thinning.

**Contribution.** In summary, our work advances literature in multiple ways: **(1) Quantifying the error of standard i.i.d. sampling:** We bring to attention and measure the approximation error introduced by using i.i.d. sampling of background and foreground data in various explanation methods. It may even lead to changes in feature importance rankings. **(2) Compress then explain (CTE):** We introduce a new paradigm of sample-efficient explainability where post-hoc explanations, like feature attributions and effects, are estimated based on a marginal distribution compressed more efficiently than with i.i.d. sampling. CTE is theoretically justified as we discover a connection between explanation estimation and distribution compression. **(3) Kernel thinning for (explainable) machine learning:** We show empirically that KT outperforms i.i.d. sampling in compressing the distribution of popular datasets used in research on explainable machine learning. In fact, this is the first work to evaluate distribution compression via KT on datasets for supervised learning, which itself is valuable. **(4) Decreasing the computational cost of explanation estimation:** We benchmark *compress then explain* (CTE) with popular explanation methods and show it results in more accurate explanations of smaller variance. CTE often achieves on-par error using 2–3× fewer samples, i.e. requiring 2–3× fewer model inferences. CTE is a simple, yet powerful, plug-in for a broad class of methods that sample from a dataset, e.g. removal-based and global explanations.

**Related work.** Our work is the first to empirically evaluate KT on datasets for supervised learning, and one of the first to reliably improve on i.i.d. sampling for multiple post-hoc explanation methods at once. Laberge et al. (2023) propose a biased sampling algorithm to attack the estimation of feature attributions, which further motivates finding robust improvements for i.i.d. sampling. Our research question is orthogonal to that of *how to sample perturbations around an input* (Petsiuk et al., 2018; Slack et al., 2021; Li et al., 2021; Ghalebikesabi et al., 2021; Li et al., 2023), or *how to efficiently sample feature coalitions* (Chen et al., 2018; Covert & Lee, 2021; Fumagalli et al., 2023). Instead of generating samples from the conditional distribution itself, which is challenging (Olsen et al., 2022), we explore how to efficiently select an appropriate subset of background data for explanations (Hase et al., 2021; Lundstrom et al., 2022). Specifically for Shapley-based explanations, Jethani et al. (2022) propose to predict them with a learned surrogate model, while Kolpaczki et al. (2024) propose their representation detached from the notion of marginal contribution. We aim to propose a general paradigm shift that benefits a broader class of explanation methods including feature effects (Apley & Zhu, 2020; Moosbauer et al., 2021) and expected gradients (Erion et al., 2021; Zhang et al., 2024).

Concerning distribution compression, the method most related to KT (Dwivedi & Mackey, 2021) is the inferior standard thinning approach (Owen, 2017). Cooper et al. (2023) use insights from KT to accelerate distributed training, while Zimmerman et al. (2024) apply KT in robotics. In the context of data-centric machine learning, we broadly relate to finding coresets to improve the efficiency of clustering (Agarwal et al., 2004; Har-Peled & Mazumdar, 2004) and active learning (Sener & Savarese, 2018), as well as dataset distillation (Wang et al., 2018) and dataset condensation (Zhao et al., 2021; Kim et al., 2022) that create synthetic samples to improve the efficiency of model training.

## 2 PRELIMINARIES

We aim to explain a prediction model trained on labeled data and denoted by $f \colon \mathcal{X} \mapsto \mathbb{R}$ where $\mathcal{X}$ is the feature space; it predicts an output using an input feature vector $\mathbf{x}$. Usually, we assume $\mathcal{X} \subseteq \mathbb{R}^d$. Without loss of generality, in the case of classification, we explain the output of a single class as a posterior probability from $[0, 1]$. We can assume a given dataset $\{(\mathbf{x}^{(1)}, y^{(1)}), \dots, (\mathbf{x}^{(n)}, y^{(n)})\}$, where every element comes from $\mathcal{X} \times \mathcal{Y}$, the underlying feature and label space, on which the explanations are computed. Depending on the explanation method and scenario, the dataset could be provided without labels. We denote such $n \times d$ dimensional dataset by $\mathbb{X}$ where $\mathbf{x}^{(i)}$ appears in the $i$-th row of $\mathbb{X}$, which is assumed to be sampled in an i.i.d. fashion from an underlying distribution $p(\mathbf{x}, y)$ defined on $\mathcal{X} \times \mathcal{Y}$. We denote a random vector as $\mathbf{X} \in \mathcal{X}$. Further, let $s \subset \{1, \dots, d\}$ be a feature index set of interest with its complement $\bar{s} = \{1, \dots, d\} \setminus s$. We index feature vectors $\mathbf{x}$ and random variables $\mathbf{X}$ by index set $s$ to restrict them to these index sets. We write $p_{\mathbf{X}}(\mathbf{x})$ and $p_{\mathbf{X}_s}(\mathbf{x}_s)$ for marginal distributions on $\mathbf{X}$ and $\mathbf{X}_s$, respectively, and $p_{\mathbf{X}_s|\mathbf{X}_t}(\mathbf{x}_s|\mathbf{x}_t)$ for conditional distribution on $\mathbf{X}_s|\mathbf{X}_t$. We use $q_{\mathbb{X}}$ to denote an empirical distribution approximating $p_{\mathbf{X}}$ based on a dataset $\mathbb{X}$.

### 2.1 SAMPLING FROM THE DATASET IS PREVALENT IN EXPLANATION ESTIMATION

Various estimators of post-hoc explanations sample from the dataset to efficiently approximate the explanation estimate (Appendix A). For example, many removal-based explanations (Covert et al., 2021) like SHAP (Lundberg & Lee, 2017) and SAGE (Covert et al., 2020) rely on marginalizing features out of the model function $f$ using their joint conditional distribution $\mathbb{E}_{\mathbf{X}_{\bar{s}} \sim p_{\mathbf{X}_{\bar{s}}|\mathbf{X}_s = \mathbf{x}_s}} [f(\mathbf{x}_s, \mathbf{X}_{\bar{s}})] = \int f(\mathbf{x}_s, \mathbf{x}_{\bar{s}}) p_{\mathbf{X}_{\bar{s}}|\mathbf{X}_s = \mathbf{x}_s}(\mathbf{x}_{\bar{s}}|\mathbf{x}_s) d\mathbf{x}_{\bar{s}}$. Note that the practical approximation of the conditional distribution $p_{\mathbf{X}_{\bar{s}}|\mathbf{X}_s = \mathbf{x}_s}(\mathbf{x}_{\bar{s}}|\mathbf{x}_s)$ itself is challenging (Chen et al., 2018; Aas et al., 2021; Olsen et al., 2022) and there is no ideal solution to this problem (see a recent benchmark by Olsen et al., 2024). For example, the default for SAGE is to assume feature independence and use the marginal distribution $p_{\mathbf{X}_{\bar{s}}|\mathbf{X}_s = \mathbf{x}_s}(\mathbf{x}_{\bar{s}}|\mathbf{x}_s) \coloneqq p_{\mathbf{X}_{\bar{s}}}(\mathbf{x}_{\bar{s}})$ (Covert et al., 2020, Appendix D); so does the KERNEL-SHAP estimator, i.e. a practical implementation of SHAP (Lundberg & Lee, 2017). This trend continues in more recent work sampling from marginal distribution (Fumagalli et al., 2023; Krzyziński et al., 2023).

**Definition 1** (Feature marginalization). *Given a set of observed values $\mathbf{x}_s$, we define a model function with marginalized features from the set $\bar{s}$ as $f(\mathbf{x}_s; p_{\mathbf{X}}) \coloneqq \mathbb{E}_{\mathbf{X}_{\bar{s}} \sim p_{\mathbf{X}_{\bar{s}}}} [f(\mathbf{x}_s, \mathbf{X}_{\bar{s}})].$*

In practice, the expectation $\mathbb{E}_{\mathbf{X}_{\bar{s}} \sim p_{\mathbf{X}_{\bar{s}}}} [f(\mathbf{x}_s, \mathbf{X}_{\bar{s}})]$ is estimated by i.i.d. sampling from the dataset $\mathbb{X}$ that approximates the distribution $p_{\mathbf{X}_{\bar{s}}}(\mathbf{x}_{\bar{s}})$. This sampled set of points forms the so-called *background data*, aka *reference points*, or *baselines* as specifically in case of the expected gradients (Erion et al., 2021) explanation method, which can be defined as EXPECTED-GRADIENTS$(\mathbf{x}) \coloneqq \mathbb{E}_{\mathbf{X} \sim p_{\mathbf{X}}, \alpha \sim U(0,1)} \left[ (\mathbf{x} - \mathbf{X}) \cdot \frac{\partial f(\mathbf{X} + \alpha \cdot (\mathbf{x} - \mathbf{X}))}{\partial \mathbf{x}} \right]$. Klein et al. (2024) benchmark feature attribution methods showing that EXPECTED-GRADIENTS is among the most faithful and robust ones.

Furthermore, i.i.d. sampling is used in global explanation methods, many of which are an *aggregation* of local explanations. To improve the computational efficiency of these approximations, often only a subset of $\mathbb{X}$ is considered, called *foreground data*. Examples include: FEATURE-EFFECTS explanations (Apley & Zhu, 2020), an aggregation of LIME (Ribeiro et al., 2016) into G-LIME (Li et al., 2023), and again SAGE, for which points from $\mathbb{X}$ require to have their corresponding labels $y$.

### 2.2 BACKGROUND ON DISTRIBUTION COMPRESSION

Standard sampling strategies can be inefficient (Dwivedi & Mackey, 2021). For example, the Monte Carlo estimate $\frac{1}{n} \sum_{i=1}^{n} h(\mathbf{x}^{(i)})$ of an unknown expectation $\mathbb{E}_{\mathbf{X} \sim p_{\mathbf{X}}} h(\mathbf{X})$ based on $n$ i.i.d. points has $\Theta(1/\sqrt{n})$ integration error $\left| \mathbb{E}_{\mathbf{X} \sim p_{\mathbf{X}}} h(\mathbf{X}) - \frac{1}{n} \sum_{i=1}^{n} h(\mathbf{x}^{(i)}) \right|$ requiring $10^2$ points for 10% relative error and $10^4$ points for 1% error (Shetty et al., 2022). To improve on i.i.d. sampling, given a sequence $\mathbb{X}$ of $n$ input points summarizing a target distribution $p_{\mathbf{X}}$, the goal of distribution compression is to identify a high quality *coreset* $\widetilde{\mathbb{X}}$ of size $\tilde{n} \ll n$. This quality is measured with the coreset's integration error $\left| \frac{1}{n} \sum_{i=1}^{n} h(\mathbf{x}^{(i)}) - \frac{1}{\tilde{n}} \sum_{i=1}^{\tilde{n}} h(\tilde{\mathbf{x}}^{(i)}) \right|$ for functions $h$ in the reproducing kernel Hilbert space induced by a given kernel function $\mathbf{k}$ (Muandet et al., 2017). The recently introduced KT algorithm (Dwivedi & Mackey, 2021; 2022) returns such a coreset that minimizes the kernel maximum mean discrepancy (MMD$_{\mathbf{k}}$, Gretton et al., 2012).

**Definition 2** (Kernel maximum mean discrepancy (Gretton et al., 2012; Dwivedi & Mackey, 2021))**.** *Let* $\mathbf{k} : \mathbb{R}^d \times \mathbb{R}^d \mapsto \mathbb{R}$ *be a bounded kernel function with* $\mathbf{k}(\mathbf{x}, \cdot)$ *measurable for all* $\mathbf{x} \in \mathbb{R}^d$, *e.g. a Gaussian kernel. Kernel maximum mean discrepancy between probability distributions* $p, q$ *on* $\mathbb{R}^d$ *is defined as* $\mathrm{MMD}_{\mathbf{k}}(p, q) \coloneqq \sup_{h \in \mathcal{H}_{\mathbf{k}}: \|h\|_{\mathbf{k}} \leq 1} \left| \mathbb{E}_{\mathbf{X} \sim p_{\mathbf{X}}} h(\mathbf{X}) - \mathbb{E}_{\mathbf{X} \sim q_{\mathbf{X}}} h(\mathbf{X}) \right|$, *where* $\mathcal{H}_{\mathbf{k}}$ *is a reproducing kernel Hilbert induced by* $\mathbf{k}$.

To formulate Propositions 1 & 2 in the next Section, we recall a biased estimate of maximum mean discrepancy, which is discussed in (Chérief-Abdellatif & Alquier, 2022, remark 3.2) and (Sriperumbudur et al., 2010, corollary 4).

**Definition 3** (Biased estimator of $\mathrm{MMD}_{\mathbf{k}}$)**.** *From (Muandet et al., 2017, section 5.1), as shown in (Gretton et al., 2012), the* $L_2$ *distance between kernel density estimates* $p_{\mathbb{X}}, q_{\mathbb{X}}$ *is a special case of the biased* $\mathrm{MMD}_{\mathbf{k}}$ *estimator, i.e. we have* $\widehat{\mathrm{MMD}}_{\mathbf{k}}^2(p_{\mathbb{X}}, q_{\mathbb{X}}) \coloneqq \|p_{\mathbb{X}} - q_{\mathbb{X}}\|_2^2 = \int \left( p_{\mathbf{X}}(\mathbf{x}) - q_{\mathbf{X}}(\mathbf{x}) \right)^2 d\mathbf{x}$.

An unbiased empirical estimate of $\mathrm{MMD}_{\mathbf{k}}$ can be relatively easily computed given a kernel function $\mathbf{k}$ (Gretton et al., 2012). COMPRESS++ (Shetty et al., 2022) is an efficient algorithm for KT that returns a coreset of size $\sqrt{n}$ in $\mathcal{O}(n \log^3 n)$ time and $\mathcal{O}(\sqrt{n} \log^2 n)$ space, making KT viable for large datasets. It was adapted to improve the kernel two-sample test (Domingo-Enrich et al., 2023).

## 3 COMPRESS THEN EXPLAIN (CTE)

We propose distribution compression as a substitute to i.i.d. sampling for feature marginalization in removal-based explanations and for aggregating global explanations. We now formalize the problem and provide theoretical intuition as to why methods for distribution compression can lead to more accurate explanation estimates. We defer the proofs to Appendix B.

**Definition 4** (Local explanation based on feature marginalization)**.** *A local explanation is a function* $g(\mathbf{x}; f, p_{\mathbf{X}}) \colon \mathcal{X} \mapsto \mathbb{R}^p$ *of input* $\mathbf{x}$ *given model* $f$ *that relies on a distribution* $p_{\mathbf{X}}$ *for feature marginalization. For estimation, it uses an empirical distribution* $q_{\mathbb{X}}$ *in place of* $p_{\mathbf{X}}$.

Examples of such local explanations include SHAP (Lundberg & Lee, 2017) and EXPECTED-GRADIENTS (Erion et al., 2021). We aim to provide high-quality explanations stemming from compressed samples as measured with a given approximation error, e.g. mean absolute error.

**Problem formulation.** To optimize the approximation error, we propose a novel formulation of the sample selection problem:

$$\min_{\widetilde{\mathbb{X}}} \quad \left\| g(\mathbf{x}; f, q_{\mathbb{X}}) - g(\mathbf{x}; f, q_{\widetilde{\mathbb{X}}}) \right\|$$
$$\text{s.t.} \quad |\widetilde{\mathbb{X}}| = \tilde{n} \ll n \tag{1}$$

for a given $\tilde{n}$, where i.i.d. sampling, distribution compression, or for example clustering, are the potential methods to find $\widetilde{\mathbb{X}}$ in an unsupervised manner. We discover a connection between distribution compression and explanation estimation in Propositions 1 & 2.

**Proposition 1** (Feature marginalization is bounded by the maximum mean discrepancy between data samples)**.** *For two empirical distributions* $q_{\mathbb{X}}, q_{\widetilde{\mathbb{X}}}$ *approximated with a kernel density estimator* $\mathbf{k}$, *we have* $\left| f(\mathbf{x}_s; q_{\mathbb{X}}) - f(\mathbf{x}_s; q_{\widetilde{\mathbb{X}}}) \right| \leq C_f \cdot \widehat{\mathrm{MMD}}_{\mathbf{k}}(q_{\mathbb{X}}, q_{\widetilde{\mathbb{X}}})$, *where* $C_f$ *denotes a constant that bounds the model function* $f$, *i.e.* $\forall_{\mathbf{x} \in \mathbb{R}^p} \left| f(\mathbf{x}) \right| \leq C_f$.

Proposition 1 provides a worst-case bound for feature marginalization, the backbone of local explanations, in terms of the $\mathrm{MMD}_{\mathbf{k}}$ distance between the (often compressed) empirical data distributions. It complements the results for input and model perturbations obtained in (Lin et al., 2023, Lemmas 1 & 4), which also shows how such a bound propagates to the local explanation function $g$. Effectively, Proposition 1 states that an algorithm minimizing $\mathrm{MMD}_{\mathbf{k}}$, e.g. KT, restricts the approximation error of explanation estimation. This makes CTE a natural contender to improve on i.i.d. sampling, given it was efficient and stable, which we evaluate empirically in extensive experiments.

**Compress then explain globally.** Local explanations are often aggregated into global explanations based on a representative sample from data, resulting in estimates of feature importance and effects.

**Definition 5** (Global explanation)**.** *A global explanation is a function that aggregates local explanations* $g$ *of model* $f$ *over input samples from distribution* $p_{\mathbf{X}}$, *i.e.* $G(p_{\mathbf{X}}; f, g) \coloneqq \mathbb{E}_{\mathbf{X} \sim p_{\mathbf{X}}} \left[ g(\mathbf{X}; f, \cdot) \right]$. *For estimation, it uses an empirical distribution* $q_{\mathbb{X}}$ *in place of* $p_{\mathbf{X}}$.

Examples of such global explanations include FEATURE-EFFECTS like partial dependence plots and accumulated local effects (Apley & Zhu, 2020), and SAGE (Covert et al., 2020), which additionally requires as an input labels $y$ of the samples drawn from $p_{\mathbf{X}}$. Notably, the local explanation function $g$ in SAGE itself relies on feature marginalization leading to using $p_{\mathbf{X}}$ twice, i.e. $\mathbb{E}_{\mathbf{X} \sim p_{\mathbf{X}}}[g(\mathbf{X}; f, p_{\mathbf{X}})]$ (see Listing 1 for a practical implementation).

**Problem formulation revisited.** For global explanations, following Equation 1, we have:

$$\min_{\widetilde{\mathbb{X}}} \quad \left\| G(q_{\mathbb{X}}; f, g) - G(q_{\widetilde{\mathbb{X}}}; f, g) \right\|$$

$$\text{s.t.} \quad |\widetilde{\mathbb{X}}| = \tilde{n} \ll n. \tag{2}$$

In Proposition 2, we conduct a worst-case analysis for global aggregated explanations.

**Proposition 2** (Global explanation is bounded by the maximum mean discrepancy between data samples). *For two empirical distributions $q_{\mathbb{X}}, q_{\widetilde{\mathbb{X}}}$ approximated with a kernel density estimator $\mathbf{k}$, we have $\left\| G(q_{\mathbb{X}}; f, g) - G(q_{\widetilde{\mathbb{X}}}; f, g) \right\|_2 \leq C_g \cdot \widehat{\mathrm{MMD}}_{\mathbf{k}}(q_{\mathbb{X}}, q_{\widetilde{\mathbb{X}}})$, where $C_g$ denotes a constant that bounds the local explanation function $g$, i.e. $\forall_{\mathbf{x} \in \mathbb{R}^p} \left\| g(\mathbf{x}; \cdot) \right\|_2 \leq C_g$.*

Analogously to Proposition 1, Proposition 2 states that an algorithm minimizing $\mathrm{MMD}_{\mathbf{k}}$, e.g. KT, restricts the approximation error of explanation estimation, which makes CTE a natural contender to improve on i.i.d. sampling. It extends the bounds for total variation distance obtained in (Baniecki et al., 2024). Moreover, in Section 4.5, we explore empirically the impact that minimizing $\mathrm{MMD}_{\mathbf{k}}$ has on decreasing alternative distribution discrepancies in practical (explainable) machine learning settings. It gives further intuition as to why clustering often leads to higher errors in explanation estimation. Our insights may guide future work on tighter theoretical guarantees for improving explanation estimation with distribution compression.

**Implementation.** The pivotal strength of CTE is that it is simple to plug into the current workflows for explanation estimation as shown in Listing 1 for SAGE. We provide analogous code listings for SHAP, EXPECTED-GRADIENTS and FEATURE-EFFECTS in Appendix C.

```python
X, y, model = ...
from goodpoints import compress
ids = compress.compresspp_kt(X, kernel_type=b"gaussian", g=4)
X_compressed = X[ids]
import sage
imputer = sage.MarginalImputer(model.predict, X_compressed)
estimator = sage.KernelEstimator(imputer)
explanation = estimator(X, y)
# or even
y_compressed = y[ids]
explanation = estimator(X_compressed, y_compressed)
```

**Listing 1:** Code snippet showing the 3-line plug-in of distribution compression for SAGE estimation.

## 4 EXPERIMENTS

In experiments, we empirically validate that the CTE paradigm improves explanation estimation across 4 methods, 2 model classes, and over 50 datasets. We compare CTE to the widely adopted practice of i.i.d. sampling (see Appendix A for further motivation). We also report sanity check results for a more deterministic baseline – sampling with k-medoids – where centroids from the clustering define a coreset from the dataset. We use the default hyperparameters of explanation algorithms (details are provided in Appendix D.1). For distribution compression, we use COMPRESS++ implemented in the `goodpoints` Python package (Dwivedi & Mackey, 2021), where we follow (Shetty et al., 2022) to use a Gaussian kernel $\mathbf{k}$ with $\sigma = \sqrt{2d}$. For all the compared methods, the subsampled set of points is of size $\sqrt{n}$ as we leave oversampling distribution compression for future work. We repeat all experiments where we apply some form of downsampling before explanation estimation 33 times and report the mean and standard error (se.) or deviation (sd.) of metric values.

**Ground truth.** The goal of CTE is to improve explanation estimation over the standard i.i.d. sampling. We measure the accuracy and effectiveness of explanation estimation with respect to a "ground truth"

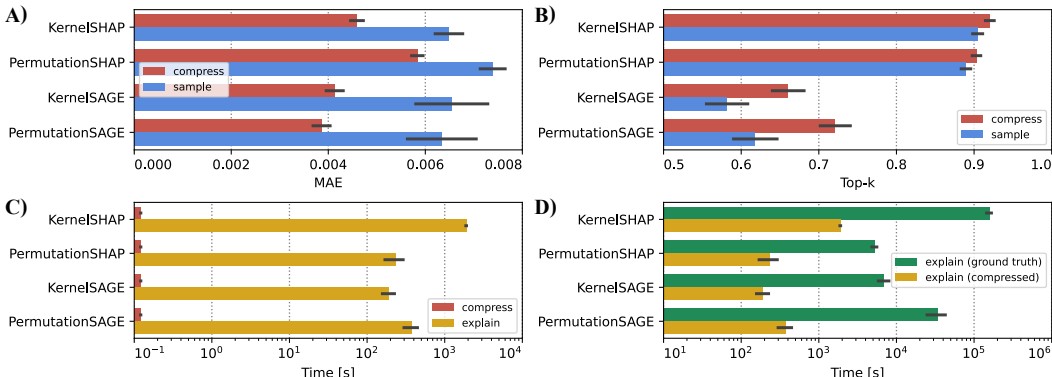

**Figure 2:** Comparison between CTE and i.i.d. sampling for the two estimators of SHAP and SAGE explanations on the `adult` dataset. **A)** We measure mean absolute error (MAE ↓) between feature attribution and importance values, and **B)** the precision in correctly identifying the 5 most important features (Top-k ↑). **C)** Comparison between the computational time of distribution compression and explanation estimation on the compressed sample, assuming the time of i.i.d. sampling is 0. **D)** Comparison between the computational time of explanation estimation on the compressed sample and on full data (in green). Analogous results for the other 4 datasets are in Appendix E. (mean ± se.)

explanation (cf. Appendix I) that is *estimated using a full validation dataset* $\mathbb{X}$, i.e. without sampling or compression. We consider settings where this is very inefficient to compute in practice ($n := n_{\text{valid}}$ is between 1000 and 25000 samples). For large datasets, we truncate the validation dataset to 20× the size of the compressed dataset. Since some explanation methods include a random component in the algorithm, we repeat their ground truth estimation 3 times and average the resulting explanations.

**Accuracy.** We are mainly interested in the accuracy of estimating a single explanation, measured by the explanation approximation error. Namely, mean absolute error (MAE), where we measure $\frac{1}{n_{\text{valid}} \cdot d} \sum_{i=1}^{n_{\text{valid}}} \left\| g(\mathbf{x}^{(i)}; f, q_{\mathbb{X}}) - g(\mathbf{x}^{(i)}; f, q_{\widetilde{\mathbb{X}}}) \right\|_1$ for SHAP and EXPECTED-GRADIENTS, and $\frac{1}{d_G} \left\| G(q_{\mathbb{X}}; f, g) - G(q_{\widetilde{\mathbb{X}}}; f, g) \right\|_1$ with $d_G = d$ for SAGE. We have $d_G = 100 \cdot (d + d^2)$ for FEATURE-EFFECTS, since we use 100 uniformly distributed grid points for 1-dimensional effects and 10×10 uniformly distributed grid points for 2-dimensional effects (see Appendix D). For broader context, in Section 4.1, we also measure the precision of correctly indicating the top $k$ features.

**Efficiency.** We measure the efficiency of compression and explanation estimation with CPU wall-clock time (in seconds), assuming the time of i.i.d. sampling is 0. We rely on popular open-source implementations of the algorithms (see Appendix C) and perform efficiency experiments on a personal computer with an M3 chip. This is to imitate the most standard workflow of explanation estimation, while we acknowledge that specific time estimates will vary in more sophisticated settings.

## 4.1 CTE IMPROVES THE ACCURACY OF ESTIMATING FEATURE ATTRIBUTIONS & IMPORTANCE

We use the preprocessed datasets and pretrained neural network models from the well-established OpenXAI benchmark (Agarwal et al., 2022). We filter out three datasets with less than 1000 observations in the validation set, where sampling is not crucial, which results in five tasks: `adult` ($n_{\text{valid}} = 9045$, $d = 13$), `compas` ($n_{\text{valid}} = 1235$, $d = 7$), `gaussian` (a synthetic dataset, $n_{\text{valid}} = 1250$, $d = 20$), `gmsc` (Give Me Some Credit, $n_{\text{valid}} = 20442$, $d = 10$), and `heloc` (aka FICO, $n_{\text{valid}} = 1975$, $d = 23$). Further details on datasets and models are provided in Appendix D.2.

We aim to show that CTE improves the estimation of feature attribution and importance explanations, namely for SHAP and SAGE. We experiment with two model-agnostic estimators: kernel-based and permutation-based. For example, Figure 2 shows the differences in MAE and Top-k between CTE and standard i.i.d. sampling on the `adult` dataset. Analogous results for the other 4 datasets from OpenXAI are shown in Appendix E. On all the considered tasks, CTE results in a notable decrease in approximation error when compared to i.i.d. sampling and an increase in precision (for top-$k$ feature identification) with negligible computational overhead. Moreover, CTE results in explanation estimates of significantly smaller variance on average as shown in Table 1.

**Table 1:** We report the improvement in the mean absolute error of estimating four popular explanations on five datasets from the OpenXAI benchmark. CTE not only improves the accuracy over i.i.d. by 20–45%, but also leads to more stable estimates by about 50%. MAE ($\downarrow$, $\pm$ sd.) values are scaled and rounded to improve readability while detailed and extended results are reported in Appendix E.

| Task | Explanation estimator | | | |
|---|---|---|---|---|
| | KERNEL-SHAP | PERMUTATION-SHAP | KERNEL-SAGE | PERMUTATION-SAGE |
| adult | $\left(\ \mathbf{i.i.d.}\ \xrightarrow{\ \text{diff.}\ }\ \mathbf{CTE}\ \right)$ | $73_{\pm14} \xrightarrow{21\%} 58_{\pm6}$ | $65_{\pm42} \xrightarrow{37\%} 41_{\pm10}$ | $63_{\pm40} \xrightarrow{40\%} 38_{\pm10}$ |
| compas | $10_{\pm4} \xrightarrow{40\%} 6_{\pm2}$ | $11_{\pm4} \xrightarrow{45\%} 6_{\pm2}$ | $29_{\pm17} \xrightarrow{38\%} 18_{\pm9}$ | $28_{\pm16} \xrightarrow{39\%} 17_{\pm8}$ |
| gaussian | $13_{\pm2} \xrightarrow{38\%} 8_{\pm1}$ | $15_{\pm2} \xrightarrow{27\%} 11_{\pm1}$ | $52_{\pm27} \xrightarrow{42\%} 30_{\pm7}$ | $52_{\pm26} \xrightarrow{44\%} 29_{\pm7}$ |
| gmsc | $23_{\pm6} \xrightarrow{39\%} 14_{\pm3}$ | $25_{\pm5} \xrightarrow{32\%} 17_{\pm3}$ | $30_{\pm13} \xrightarrow{40\%} 18_{\pm5}$ | $28_{\pm14} \xrightarrow{43\%} 16_{\pm5}$ |
| heloc | $67_{\pm15} \xrightarrow{39\%} 41_{\pm7}$ | $72_{\pm13} \xrightarrow{33\%} 48_{\pm6}$ | $34_{\pm10} \xrightarrow{21\%} 27_{\pm6}$ | $29_{\pm11} \xrightarrow{28\%} 21_{\pm6}$ |

## 4.2 CTE AS AN EFFICIENT ALTERNATIVE TO I.I.D. SAMPLING IN EXPLANATION ESTIMATION

We find CTE to be a very efficient alternative to standard i.i.d. sampling in explanation estimation. For example, compressing a distribution from 1k to 32 samples takes less than 0.1 seconds, and from 20k to 128 samples takes less than 1 second. The exact runtime will, of course, differ based on the number of features. Figure 3 reports the wall-clock time for datasets of different sizes. Note that the potential runtimes for distribution compression are of magnitudes smaller than the typical runtime of explanation estimation. For example, estimating KERNEL-SHAP for 9k samples using 128 background samples takes 30 minutes, which is about $60\times$ less than estimating the ground truth explanation (Figure 2). Moreover, estimating KERNEL-SHAP or PERMUTATION-SAGE for 1k samples using 32 background samples takes about 10 seconds, which is about $30\times$ less than estimating the ground truth explanation (Appendix E).

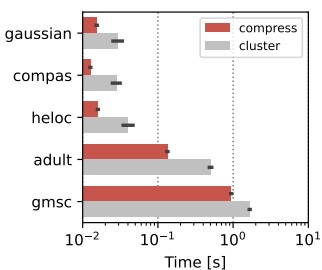

**Figure 3:** Compressing a distribution from 20k to 128 samples takes less than 1 second to compute on a CPU. (mean $\pm$ se.)

## 4.3 CTE IMPROVES GRADIENT-BASED EXPLANATIONS SPECIFIC TO NEURAL NETWORKS

Next, we aim to show the broader applicability of CTE by evaluating it on gradient-based explanations specific to neural networks, often fitted to larger unstructured datasets.

**Sanity check.** We first compress the validation sets of IMDB and ImageNet-1k on a single CPU as a sanity check for the viability of CTE in settings considering larger datasets. For the IMDB dataset ($n_{\text{valid}} = 25000$, $d = 768$), CTE takes as an input text embeddings from the pretrained DistilBERT model's last layer (preceding a classifier) that has a dimension of size 768. Similarly, for ImageNet-1k ($n_{\text{valid}} = 50000$, $d = 512$), CTE operates on the hidden representation extracted from ResNet-18. Figure 4 shows the optimized $\text{MMD}_{\mathbf{k}}$ metric between the distributions and computation time in seconds. We can see that proper compression results in huge benefits w.r.t. $\text{MMD}_{\mathbf{k}}$ (compared to i.i.d. sampling and clustering) and only negligible computational overhead.

**Accuracy and efficiency.** We now study CTE together with EXPECTED-GRADIENTS of neural network models trained to 18 datasets ($n_{\text{valid}} > 1000$, $d \geq 32$) from the OpenML-CC18 (Bischl et al., 2021) and OpenML-CTR23 (Fischer et al., 2023) benchmark suites. Details on datasets and models are provided in Appendix D.2.

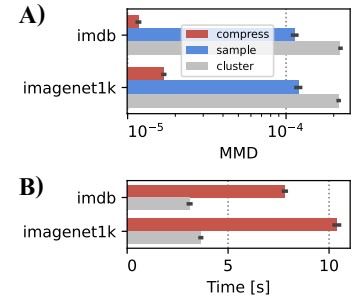

**Figure 4:** **A)** COMPRESS++ effectively optimizes $\text{MMD}_{\mathbf{k}}$ on unstructured IMDB and Imagenet-1k datasets. **B)** Compressing a distribution from 25k–50k to 128 samples in 512–768 dimensions takes about 5–10 sec. to compute on a CPU. (mean $\pm$ se.)

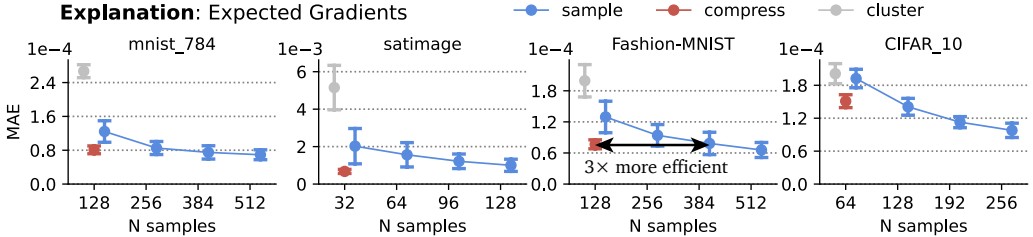

**Figure 5:** Comparison between CTE, i.i.d. sampling and clustering for EXPECTED-GRADIENTS explanations on the 4 image classification datasets. We measure mean absolute error (MAE ↓) between feature attribution values. CTE is not only more accurate but also more stable as measured with deviation. Analogous results for the remaining 14 datasets are in Appendix F. (mean ± sd.)

In Figure 5, we show the explanation approximation error for 4 image classification tasks, while analogous results for the remaining 14 datasets are provided in Appendix F. Additionally here, we vary the number of data points sampled from i.i.d. to inspect the increase in efficiency of CTE. In all cases, CTE achieves on-par approximation error using fewer samples than i.i.d. sampling, i.e. requiring fewer model inferences, resulting in faster computation and saved resources. The accuracy improvements are significant, i.e. CTE decreases the estimation error for EXPECTED-GRADIENTS by 35% on mnist_784 (Welch's t-test: $p < 1e{-}10$), by 40% on Fashion-MNIST ($p < 1e{-}10$), and by 21% on CIFAR_10 ($p < 1e{-}10$). Moreover, CTE provides 2–3× efficiency improvements as measured by the number of samples required for i.i.d. to reach the error of CTE.

**Model-agnostic explanation of a language model.** In Appendix F, we further experiment with applying CTE to improve the estimation of G-LIME (Li et al., 2023) explaining the predictions of a DistilBERT language model trained on the IMDB dataset for sentiment analysis.

### 4.4 ABLATIONS WITH ANOTHER 30 DATASETS, AN XGBOOST MODEL & FEATURE EFFECTS

For a convincing case to use CTE instead of i.i.d. in practice, we perform additional empirical analysis on various datasets, with a different model class, and include another global explanation method. More specifically, we use CTE to improve FEATURE-EFFECTS of XGBoost models trained on further 30 datasets ($n_{valid} > 1000$, $d < 32$) from OpenML-CC18 and OpenML-CTR23. Details on datasets and models are provided in Appendix D.2. Moreover, we include SHAP and SAGE in the benchmark similarly to Section 4.1. As another ablation, SAGE is evaluated in two variants that consider either compressing only the background data (a rather typical scenario), or using the compressed samples as both background and foreground data (as indicated with "fg."; refer to Listing 1 for this distinction).

Figure 6 shows the explanation approximation error for 3 predictive tasks, while analogous results for the remaining explanation estimators and 27 datasets are provided in Appendix G. We observe that CTE significantly improves the estimation of FEATURE-EFFECTS in all cases. We further confirm the conclusions from Sections 4.1 & 4.2 that CTE improves SHAP and SAGE. Another insight is that, on average, CTE provides a smaller improvement over i.i.d. sampling when considering compressing foreground data in SAGE.

**Conclusion from the experiments.** In Figure 7, we aggregated the results from Sections 4.3 & 4.4 to conclude the main claim that CTE offers 2–3× improvements in efficiency over i.i.d. sampling.

### 4.5 KERNEL THINNING ON DATASETS FOR (EXPLAINABLE) MACHINE LEARNING

We have already established that distribution compression is a viable approach to data sampling, which regularly entails a better approximation of explanations. Moreover, its computational overhead is negligible when applied before explanation estimation. To provide more context on discovering the theoretical justification for CTE from Section 3, we aim to show that COMPRESS++ entails a better approximation of feature distribution on popular datasets for (explainable) machine learning. This is out-of-the-box, without tuning its hyperparameters, which is a natural direction for future work.

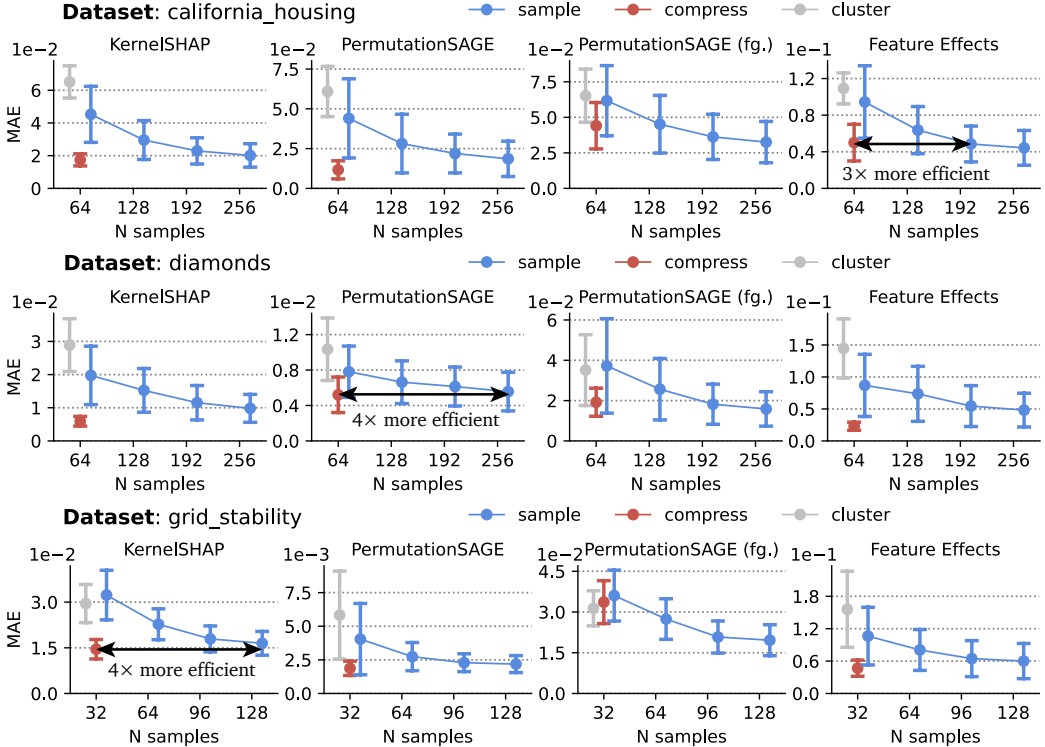

**Figure 6:** CTE improves the approximation error of local and global removal-based explanations. SAGE is evaluated in two variants that consider either compressing only the background data (default), or using the compressed samples as both background and foreground data (as indicated with "fg."). Analogous results for the remaining estimators and 27 datasets are in Appendix G. (mean ± sd.)

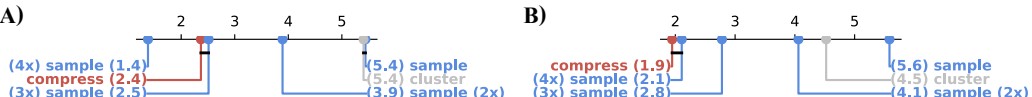

**Figure 7:** Critical difference diagrams of average ranks (lower is better) aggregated over 6 explanation estimators and 48 dataset–model pairs: **A)** for MAE averaged over repeats, and **B)** for the sd. of MAE over repeats that corresponds to the stability of explanation estimation. CTE often achieves on-par explanation approximation error using 2–3× fewer samples, i.e. requiring 2–3× fewer model inferences, which is efficient. Moreover, CTE guarantees more stable estimates than i.i.d. sampling.

**Measuring distribution change.** In general, measuring the similarity of distributions or datasets is challenging, and many metrics with various properties have been proposed for this task (Gibbs & Su, 2002). Here, we report the following distance metric values between the original and compressed distribution: the optimized $\mathrm{MMD_k}$, total variation distance (TV, Gibbs & Su, 2002), Kullback–Leibler divergence (KL, Gibbs & Su, 2002), and d-dimensional Wasserstein distance (WD, Feydy et al., 2019; Laberge et al., 2023). Since approximating d-dimensional TV and KL is infeasible in practice (see e.g. Sriperumbudur et al., 2012, section 5), we report an average of the top-3 largest discrepancies between the 1-dimensional distributions of features as a proxy.

**Result.** In Figure 8, we observe that COMPRESS++ works much better in terms of $\mathrm{MMD_k}$ on all five datasets from OpenXAI, compared to standard i.i.d. sampling or the clustering baseline, which is no surprise as this metric is internally optimized by the former. Note that it also leads to notable improvements in all other metrics. Overall, there is no consistent improvement in approximating the distribution using clustering, which explains why it leads to higher error in explanation estimation.

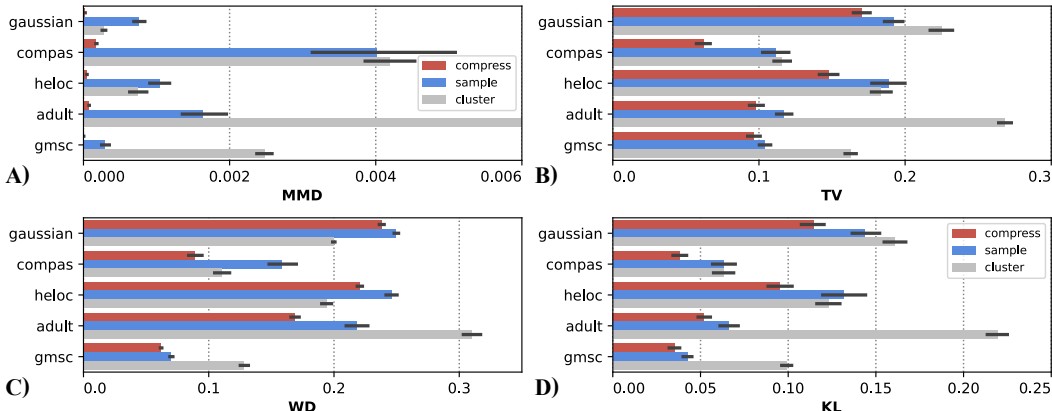

**Figure 8:** COMPRESS++ with Gaussian kernel on five datasets (rows) for the four considered distribution change metrics (panels): **A)** $\mathrm{MMD}_\mathbf{k}$, **B)** total variation distance, **C)** d-dimensional Wasserstein distance, and **D)** Kullback–Leibler divergence. The length of the bar is the mean value $\pm$ standard error across statistical repetitions, where color indicates the applied downsampling method.

## 5 DISCUSSION

We propose *compress then explain* as a powerful alternative to the conventional *sample then explain* paradigm in explanation estimation. CTE has the potential to improve approximation error across a wide range of explanation methods for various predictive tasks. Specifically, we show accuracy and stability improvements in popular removal-based explanations that marginalize feature influence, and in general, global explanations that aggregate local explanations over a subset of data. Moreover, CTE leads to more efficient explanation estimation by decreasing the computational resources (time, model inferences) required to achieve error on par with a larger i.i.d. sample size.

**Future work** on methods for marginal distribution compression other than kernel thinning and clustering will bring further improvements in the performance of explanation estimation. Distribution compression methods, by design, have hyperparameters that may impact the empirical results. Although we have shown that the default COMPRESS++ algorithm is a robust baseline, exploring the tunability of its hyperparameters is a natural future work direction (similarly as in the case of conditional sampling methods, Olsen et al., 2024). We used the Gaussian kernel because it is the standard in the field of distribution compression (Dwivedi & Mackey, 2022; Shetty et al., 2022; Domingo-Enrich et al., 2023), especially in experimental analysis, and is generally adopted within machine learning applications. Although our empirical validation shows that the Gaussian kernel works well for over 50 datasets, exploring other kernels for which theoretical thinning error bounds exist, like Matérn or B-spline (Dwivedi & Mackey, 2021), is a viable future work direction. Furthermore, especially for tabular datasets, dealing with categorical features can be an issue, which we elaborate on in Appendix D.2. Specifically for supervised learning, a stratified variant of kernel thinning taking into account a distribution of the target feature could further improve loss-based explanations like SAGE, or even the estimation of group fairness metrics. In concurrent work, Gong et al. (2024) generalize KT to speed up supervised learning problems involving kernel methods. One could also investigate how influence functions (Koh & Liang, 2017), which aim to attribute the importance of data to the model's prediction, can guide sampling for efficient explanation estimation.

**Broader impact.** In general, improving explanation methods has positive implications for humans interacting with AI systems (Rong et al., 2024). But, specifically in the context of this work, biased sampling can be exploited to manipulate the explanation results (Slack et al., 2020; Baniecki & Biecek, 2022; Laberge et al., 2023). CTE could minimize the risk of such adversaries and prevent "random seed/state hacking" based on the rather unstable i.i.d. sampling from data in empirical research (Herrmann et al., 2024).

**Code.** We provide additional details on reproducibility in the Appendix, as well as the code to reproduce all experiments in this paper is available at `https://github.com/hbaniecki/compress-then-explain`.

## ACKNOWLEDGMENTS

This work was financially supported by the Polish National Science Centre grant number 2021/43/O/ST6/00347, and carried out with the support of the Laboratory of Bioinformatics and Computational Genomics and the High Performance Computing Center of the Faculty of Mathematics and Information Science, Warsaw University of Technology. Hubert Baniecki gratefully acknowledges scholarship funding from the Polish National Agency for Academic Exchange under the Preludium Bis NAWA 3 program. We want to thank Maximilian Muschalik, Mateusz Biesiadowski, Paulina Kaczynska, Anna Semik, and anonymous reviewers for their valuable feedback regarding this work.

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

APPENDIX FOR "EFFICIENT AND ACCURATE EXPLANATION ESTIMATION WITH DISTRIBUTION COMPRESSION"

In Appendix B, we derive proofs for Propositions 1 & 2. Appendix C provides code listings for SHAP, EXPECTED-GRADIENTS and FEATURE-EFFECTS, analogous to Listing 1 for SAGE. Additional details on the experimental setup are provided in Appendix D. Appendices E, F & G report experimental results for the remaining datasets. Appendix H comments on compute resources used for experiments. Appendix I provides exemplary visual comparisons of explanations. The code to reproduce all experiments in this paper is available at `https://github.com/hbaniecki/compress-then-explain`.

## A    MOTIVATION: STANDARD I.I.D. SAMPLING IN EXPLANATION ESTIMATION

We find that i.i.d. sampling from datasets is a heuristic often used (and overlooked) in various estimators of post-hoc explanations. Our work aims to first quantify the approximation error introduced by *sample then explain*, and then propose a method to efficiently reduce it. Below are a few examples from the literature on explainability that motivate the shift to our introduced *compress then explain* paradigm.

In (Laberge et al., 2023), we read "*For instance, when a dataset is used to represent a background distribution, explainers in the SHAP library such as the ExactExplainer and TreeExplainer will subsample this dataset by selecting 100 instances uniformly at random when the size of the dataset exceeds 100.*"

In (Chen & Sun, 2023), we read "*[Footnote 1.] We use a random subset of samples for each class in the real implementation, to reduce the computation costs of clustering.*"

In (Ghalebikesabi et al., 2021), we read "*After training a convolutional neural network on the MNIST dataset, we explain digits with the predicted label 8 given a background dataset of 100 images with labels 3 and 8.*", as well as "*Feature attributions are sorted by similarity according to a preliminary PCA analysis across a subset of 2000 samples from the Adult Income dataset, using 2000 reference points.*"

In (Erion et al., 2021), we read "*During training, we let k be the number of samples we draw to compute expected gradients for each mini-batch.* and "*This expectation-based formulation lends itself to a natural, sampling based approximation method: (1) draw samples of $x'$ from the training dataset [...], (2) compute the value inside the expectation for each sample and (3) average over samples.*"

In (Van Looveren & Klaise, 2021), we read "*We also need a representative, unlabeled sample of the training dataset.*", and in Algorithms 1 and 2: "*A sample $X = \{x_1, \ldots, x_n\}$ from training set.*"

In (Covert et al., 2020), we read "*When calculating feature importance, our sampling approximation for SAGE (Algorithm 1) was run using draws from the marginal distribution. We used a fixed set of 512 background samples for the bank, bike and credit datasets, 128 for MNIST, and all 334 training examples for BRCA.*"

In the shap Python package (Lundberg & Lee, 2017), there is a warning saying "*Using 110 background data samples could cause slower run times. Consider using shap.sample(data, K) or shap.kmeans(data, K) to summarize the background as K samples.*", and the documentation mentions "*For small problems, this background dataset can be the whole training set, but for larger problems consider using a single reference value or using the **kmeans** function to summarize the dataset.*"

# B PROOFS

Below, we derive proofs for Propositions 1 & 2.

**Proposition 1** (Feature marginalization is bounded by the maximum mean discrepancy between data samples). *For two empirical distributions $q_{\mathbb{X}}, q_{\widetilde{\mathbb{X}}}$ approximated with a kernel density estimator $\mathbf{k}$, we have $\left| f(\mathbf{x}_s; q_{\mathbb{X}}) - f(\mathbf{x}_s; q_{\widetilde{\mathbb{X}}}) \right| \leq C_f \cdot \widehat{\mathrm{MMD}}_{\mathbf{k}}(q_{\mathbb{X}}, q_{\widetilde{\mathbb{X}}})$, where $C_f$ denotes a constant that bounds the model function $f$, i.e. $\forall_{\mathbf{x} \in \mathbb{R}^p} \left| f(\mathbf{x}) \right| \leq C_f$.*

*Proof.* We derive the following inequality:

$$\left( f(\mathbf{x}_s; p_{\mathbf{X}_{\bar{s}}}) - f(\mathbf{x}_s; q_{\mathbf{X}_{\bar{s}}}) \right)^2 = \left( \mathbb{E}_{\mathbf{X}_{\bar{s}} \sim p_{\mathbf{X}_{\bar{s}}}} [f(\mathbf{x}_s, \mathbf{X}_{\bar{s}})] - \mathbb{E}_{\mathbf{X}_{\bar{s}} \sim q_{\mathbf{X}_{\bar{s}}}} [f(\mathbf{x}_s, \mathbf{X}_{\bar{s}})] \right)^2 \tag{3}$$

$$= \left( \int f(\mathbf{x}_s, \mathbf{x}_{\bar{s}}) p_{\mathbf{X}_{\bar{s}}}(\mathbf{x}_{\bar{s}}) d\mathbf{x}_{\bar{s}} - \int f(\mathbf{x}_s, \mathbf{x}_{\bar{s}}) q_{\mathbf{X}_{\bar{s}}}(\mathbf{x}_{\bar{s}}) d\mathbf{x}_{\bar{s}} \right)^2 \tag{4}$$

$$(\text{linearity}) = \left( \int f(\mathbf{x}_s, \mathbf{x}_{\bar{s}}) \left( p_{\mathbf{X}_{\bar{s}}}(\mathbf{x}_{\bar{s}}) - q_{\mathbf{X}_{\bar{s}}}(\mathbf{x}_{\bar{s}}) \right) d\mathbf{x}_{\bar{s}} \right)^2 \tag{5}$$

$$(\text{Cauchy–Schwarz}) \leq \int \left( f(\mathbf{x}_s, \mathbf{x}_{\bar{s}}) \right)^2 d\mathbf{x}_{\bar{s}} \int \left( p_{\mathbf{X}_{\bar{s}}}(\mathbf{x}_{\bar{s}}) - q_{\mathbf{X}_{\bar{s}}}(\mathbf{x}_{\bar{s}}) \right)^2 d\mathbf{x}_{\bar{s}} \tag{6}$$

$$(\text{boundedness}) \leq C_f^2 \cdot \int \left( p_{\mathbf{X}_{\bar{s}}}(\mathbf{x}_{\bar{s}}) - q_{\mathbf{X}_{\bar{s}}}(\mathbf{x}_{\bar{s}}) \right)^2 d\mathbf{x}_{\bar{s}} \tag{7}$$

$$(\text{Definition 3}) = C_f^2 \cdot \widehat{\mathrm{MMD}}_{\mathbf{k}}^2(p_{\mathbf{X}_{\bar{s}}}, q_{\mathbf{X}_{\bar{s}}}). \tag{8}$$

Substituting with empirical distributions, we have

$$\left| f(\mathbf{x}_s; q_{\mathbb{X}}) - f(\mathbf{x}_s; q_{\widetilde{\mathbb{X}}}) \right| \leq C_f \cdot \widehat{\mathrm{MMD}}_{\mathbf{k}}(q_{\mathbb{X}}, q_{\widetilde{\mathbb{X}}}). \tag{9}$$

$\square$

**Proposition 2** (Global explanation is bounded by the maximum mean discrepancy between data samples). *For two empirical distributions $q_{\mathbb{X}}, q_{\widetilde{\mathbb{X}}}$ approximated with a kernel density estimator $\mathbf{k}$, we have $\left\| G(q_{\mathbb{X}}; f, g) - G(q_{\widetilde{\mathbb{X}}}; f, g) \right\|_2 \leq C_g \cdot \widehat{\mathrm{MMD}}_{\mathbf{k}}(q_{\mathbb{X}}, q_{\widetilde{\mathbb{X}}})$, where $C_g$ denotes a constant that bounds the local explanation function $g$, i.e. $\forall_{\mathbf{x} \in \mathbb{R}^p} \left\| g(\mathbf{x}; \cdot) \right\|_2 \leq C_g$.*

*Proof.* We derive the following inequality:

$$\left\| G(p_{\mathbf{X}}; f, g) - G(q_{\mathbf{X}}; f, g) \right\|_2^2 = \left\| \mathbb{E}_{\mathbf{X} \sim p_{\mathbf{X}}} [g(\mathbf{X}; f, \cdot)] - \mathbb{E}_{\mathbf{X} \sim q_{\mathbf{X}}} [g(\mathbf{X}; f, \cdot)] \right\|_2^2 \tag{10}$$

$$= \left\| \int g(\mathbf{x}) p_{\mathbf{X}}(\mathbf{x}) d\mathbf{x} - \int g(\mathbf{x}) q_{\mathbf{X}}(\mathbf{x}) d\mathbf{x} \right\|_2^2 \tag{11}$$

$$(\text{linearity}) = \left\| \int g(\mathbf{x}) \left( p_{\mathbf{X}}(\mathbf{x}) - q_{\mathbf{X}}(\mathbf{x}) \right) d\mathbf{x} \right\|_2^2 \tag{12}$$

$$(\text{Cauchy–Schwarz}) \leq \int \left\| g(\mathbf{x}) \right\|_2^2 d\mathbf{x} \int \left( p_{\mathbf{X}}(\mathbf{x}) - q_{\mathbf{X}}(\mathbf{x}) \right)^2 d\mathbf{x} \tag{13}$$

$$(\text{boundedness}) \leq C_g^2 \cdot \int \left( p_{\mathbf{X}}(\mathbf{x}) - q_{\mathbf{X}}(\mathbf{x}) \right)^2 d\mathbf{x} \tag{14}$$

$$(\text{Definition 3}) = C_g^2 \cdot \widehat{\mathrm{MMD}}_{\mathbf{k}}^2(p_{\mathbf{X}}, q_{\mathbf{X}}). \tag{15}$$

Substituting with empirical distributions, we have

$$\left\| G(q_{\mathbb{X}}; f, g) - G(q_{\widetilde{\mathbb{X}}}; f, g) \right\|_2 \leq C_g \cdot \widehat{\mathrm{MMD}}_{\mathbf{k}}(q_{\mathbb{X}}, q_{\widetilde{\mathbb{X}}}). \tag{16}$$

$\square$

## C IMPLEMENTING CTE IN PRACTICE

CTE is simple to plug-into the current workflows for explanation estimation as shown in Listing 1 for SAGE, Listing 2 for SHAP, Listing 3 for EXPECTED-GRADIENTS, and Listing 4 for FEATURE-EFFECTS. We use the `goodpoints` Python package (Dwivedi & Mackey, 2021, MIT license).

```python
X, model = ...
from goodpoints import compress
ids = compress.compresspp_kt(X, kernel_type=b"gaussian", g=4)
X_compressed = X[ids]
import shap
masker = shap.maskers.Independent(X_compressed)
explainer = shap.PermutationExplainer(model.predict, masker)
explanation = explainer(X)
```

**Listing 2:** Code snippet showing the 3-line plug-in of distribution compression for SHAP estimation.

```python
X, model = ...
from goodpoints import compress
ids = compress.compresspp_kt(X, kernel_type=b"gaussian", g=4)
X_compressed = X[ids]
import captum
explainer = captum.attr.IntegratedGradients(model)
import torch
inputs = torch.as_tensor(X)
baselines = torch.as_tensor(X_compressed)
explanation = torch.mean(torch.stack([
    explainer.attribute(inputs, baselines[[i]], target=1)
    for i in range(baselines.shape[0])
]), dim=0)
```

**Listing 3:** Code snippet showing the plug-in of distribution compression for EXPECTED-GRADIENTS.

```python
X, model = ...
from goodpoints import compress
ids = compress.compresspp_kt(X, kernel_type=b"gaussian", g=4)
X_compressed = X[ids]
import alibi
explainer = alibi.explainers.PartialDependence(predictor=model.predict)
explanation = explainer.explain(X_compressed)
```

**Listing 4:** Code snippet showing the plug-in of distribution compression for FEATURE-EFFECTS.

# D  EXPERIMENTAL SETUP

## D.1  EXPLANATION HYPERPARAMETERS

In Section 4, we experiment with 4 explanation methods (6 estimators). Without the loss of generality, in case of classification models, we always explain a prediction for the 2nd class. For SHAP, we use the KERNEL-SHAP and PERMUTATION-SHAP implementations from the `shap` Python package (Lundberg & Lee, 2017, MIT license) with default hyperparameters (notably, `npermutations=10` in the latter). For SAGE, we use the KERNEL-SAGE and PERMUTATION-SAGE implementations from the `sage` Python package (Covert et al., 2020, MIT license). We use default hyperparameters; notably, a cross-entropy loss for classification and mean squared error for regression. For EXPECTED-GRADIENTS, we aggregate with mean the integrated gradients explanations from the `captum` Python package (Kokhlikyan et al., 2020, BSD-3 license), for which we use default hyperparameters; notably, `n_steps=50` and `method="gausslegendre"`. For FEATURE-EFFECTS, we implement the partial dependence algorithm (Apley & Zhu, 2020; Moosbauer et al., 2021) ourselves for maximum computational speed in case of 2-dimensional plots, mimicking the popular open-source implementations.[1] We use 100 uniformly distributed grid points for 1-dimensional plots and 10×10 uniformly distributed grid points for 2-dimensional plots.

## D.2  DETAILS ON DATASETS AND MODELS

Table 2 shows details of datasets from the OpenXAI (Agarwal et al., 2022, MIT license) benchmark used in Sections 4.1, 4.2 & 4.5. To each dataset, there is a pretrained neural network with an accuracy of 92% (`gaussian`), 85% (`compas`), 74% (`heloc`), 85% (`adult`) and 93% (`gmsc`). We do not further preprocess data; notably, feature values are already scaled to $[0, 1]$.

**Table 2:** Datasets from OpenXAI with $n_{\text{valid}} > 1000$ used in experiments.

| Dataset | $n_{\text{train}}$ | $n_{\text{valid}}$ | $d$ | No. classes |
|---------|-------|-------|----|-------------|
| gaussian | 3750 | 1250 | 20 | 2 |
| compas | 4937 | 1235 | 7 | 2 |
| heloc | 7896 | 1975 | 23 | 2 |
| adult | 36177 | 9045 | 13 | 2 |
| gmsc | 81767 | 20442 | 10 | 2 |

Table 3 shows details of datasets from the OpenML-CC18 (Bischl et al., 2021, BSD-3 license) and OpenML-CTR23 (Fischer et al., 2023, BSD-3 license) benchmarks used in Sections 4.3 & 4.4. We first split all datasets in 75:25 (train:validation) ratio and left 48 datasets with $n_{\text{valid}} > 1000$ for our experiments. For the 30 smaller ($d < 32$) datasets, we train an XGBoost model with default hyperparameters (200 estimators) and explain it with SHAP, SAGE, FEATURE-EFFECTS. For the 18 bigger ($d \geq 32$) datasets, we train a 3-layer neural network model with (128, 64) neurons in hidden ReLU layers and explain it with EXPECTED-GRADIENTS. We perform basic preprocessing of data: (1) remove features with a single or $n$ unique values, (2) target encode categorical features, (3) impute missing values with mean, and (4) standardize features.

In general, categorical features can be an issue for clustering and distribution compression algorithms; so are for many explanation algorithms and conditional distribution samplers. Although target encoding worked well in our setup, we envision two additional heuristics to deal with categorical features: (1) perform distribution compression using a dataset restricted to non-categorical features, (2) target encode categorical features only for distribution compression.

---

[1] https://docs.seldon.io/projects/alibi/en/latest/api/alibi.explainers.html#alibi.explainers.PartialDependence; https://interpret.ml/docs/python/api/PartialDependence

**Table 3:** Datasets from OpenML-CC18 and OpenML-CTR23 with $n_{\text{valid}} > 1000$ used in experiments.

| Dataset | Task ID | $n_{\text{train}}$ | $n_{\text{valid}}$ | $d$ | No. classes |
|---|---|---|---|---|---|
| phoneme | 9952 | 4053 | 1351 | 5 | 2 |
| wilt | 146820 | 3629 | 1210 | 5 | 2 |
| cps88wages | 361261 | 21116 | 7039 | 6 | – |
| jungle_chess | 167119 | 33614 | 11205 | 6 | 3 |
| abalone | 361234 | 3132 | 1045 | 8 | – |
| electricity | 219 | 33984 | 11328 | 8 | 2 |
| kin8nm | 361258 | 6144 | 2048 | 8 | – |
| california_housing | 361255 | 15480 | 5160 | 8 | – |
| brazilian_houses | 361267 | 8019 | 2673 | 9 | – |
| diamonds | 361257 | 40455 | 13485 | 9 | – |
| physiochemical_protein | 361241 | 34297 | 11433 | 9 | – |
| white_wine | 361249 | 3673 | 1225 | 11 | – |
| health_insurance | 361269 | 16704 | 5568 | 11 | – |
| grid_stability | 361251 | 7500 | 2500 | 12 | – |
| adult | 7592 | 36631 | 12211 | 14 | 2 |
| naval_propulsion_plant | 361247 | 8950 | 2984 | 14 | – |
| miami_housing | 361260 | 10449 | 3483 | 15 | – |
| letter | 6 | 15000 | 5000 | 16 | 26 |
| bank-marketing | 14965 | 33908 | 11303 | 16 | 2 |
| pendigits | 32 | 8244 | 2748 | 16 | 10 |
| video_transcoding | 361252 | 51588 | 17196 | 18 | – |
| churn | 167141 | 3750 | 1250 | 20 | 2 |
| kings_county | 361266 | 16209 | 5404 | 21 | – |
| numerai28.6 | 167120 | 72240 | 24080 | 21 | 2 |
| sarcos | 361254 | 36699 | 12234 | 21 | – |
| cpu_activity | 361256 | 6144 | 2048 | 21 | – |
| jm1 | 3904 | 8163 | 2722 | 21 | 2 |
| wall-robot-navigation | 9960 | 4092 | 1364 | 24 | 4 |
| fifa | 361272 | 14383 | 4795 | 28 | – |
| PhishingWebsites | 14952 | 8291 | 2764 | 30 | 2 |
| pumadyn32nh | 361259 | 6144 | 2048 | 32 | – |
| GestureSegmentation | 14969 | 7404 | 2469 | 32 | 5 |
| satimage | 2074 | 4822 | 1608 | 36 | 6 |
| texture | 125922 | 4125 | 1375 | 40 | 11 |
| connect-4 | 146195 | 50667 | 16890 | 42 | 3 |
| fps_benchmark | 361268 | 18468 | 6156 | 43 | – |
| wave_energy | 361253 | 54000 | 18000 | 48 | – |
| theorem-proving | 9985 | 4588 | 1530 | 51 | 6 |
| spambase | 43 | 3450 | 1151 | 57 | 2 |
| optdigits | 28 | 4215 | 1405 | 64 | 10 |
| superconductivity | 361242 | 15947 | 5316 | 81 | – |
| nomao | 9977 | 25848 | 8617 | 118 | 2 |
| har | 14970 | 7724 | 2575 | 561 | 6 |
| isolet | 3481 | 5847 | 1950 | 617 | 26 |
| mnist_784 | 3573 | 52500 | 17500 | 784 | 10 |
| Fashion-MNIST | 146825 | 52500 | 17500 | 784 | 10 |
| Devnagari-Script | 167121 | 69000 | 23000 | 1024 | 46 |
| CIFAR_10 | 167124 | 45000 | 15000 | 3072 | 10 |

# E  CTE SIGNIFICANTLY IMPROVES THE ESTIMATION OF FEATURE ATTRIBUTIONS & IMPORTANCE

We report the differences in MAE and Top-k between CTE and i.i.d. sampling in Figure 9 (`compas`), Figure 10 (`heloc`), Figure 11 (`gmsc`) and Figure 12 (`gaussian`). On all the considered tasks, CTE offers a notable decrease in approximation error of SHAP and SAGE with negligible computational overhead (as measured by time in seconds).

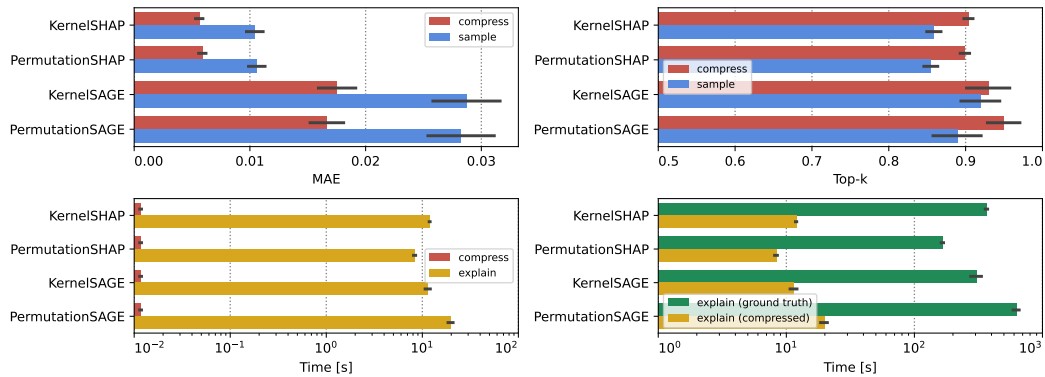

**Figure 9:** Extended Figure 2 (1/4). CTE improves SHAP and SAGE estimation by using the compressed samples as background data for the `compas` dataset. We measure mean absolute error (MAE ↓) between feature attribution and importance values, as well as the precision in correctly indicating the 3 most important features (Top-k ↑). Computational resources required to compress a distribution are negligible in the context of explanation estimation. (mean ± se.)

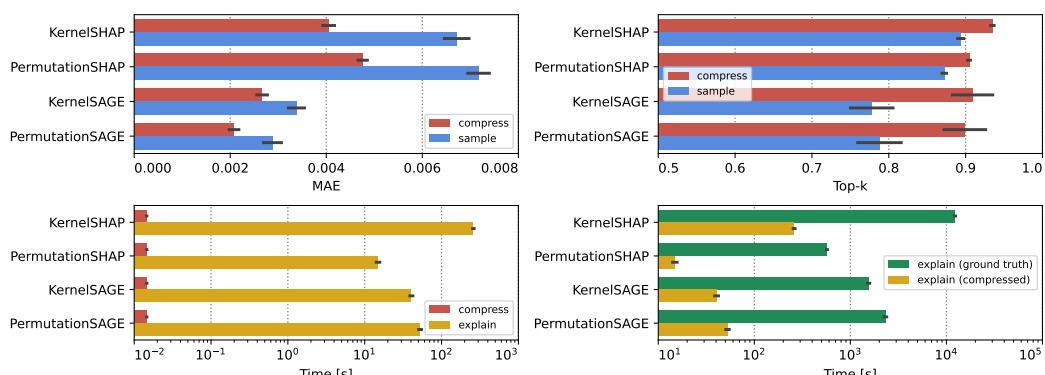

**Figure 10:** Extended Figure 2 (2/4). CTE improves SHAP and SAGE estimation on the `heloc` dataset.

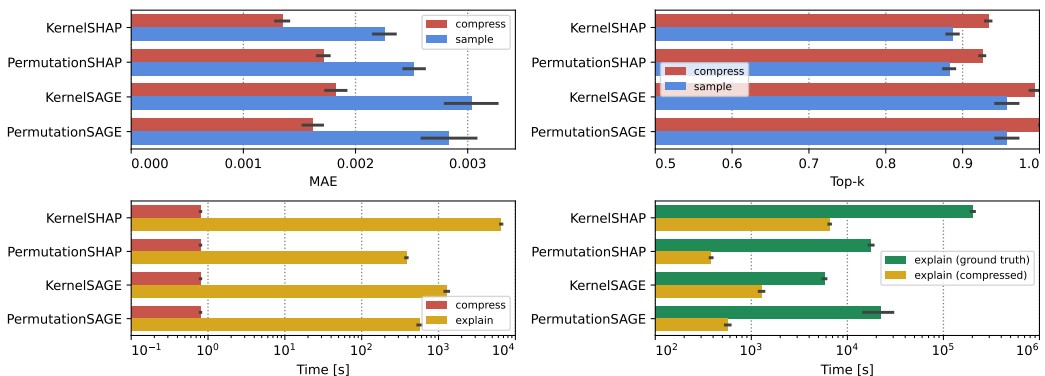

**Figure 11:** Extended Figure 2 (3/4). CTE improves SHAP and SAGE estimation by using the compressed samples as background data for the gmsc dataset. We measure mean absolute error (MAE ↓) between feature attribution and importance values, as well as the precision in correctly indicating the 5 most important features (Top-k ↑). Computational resources required to compress a distribution are negligible in the context of explanation estimation. (mean ± se.)

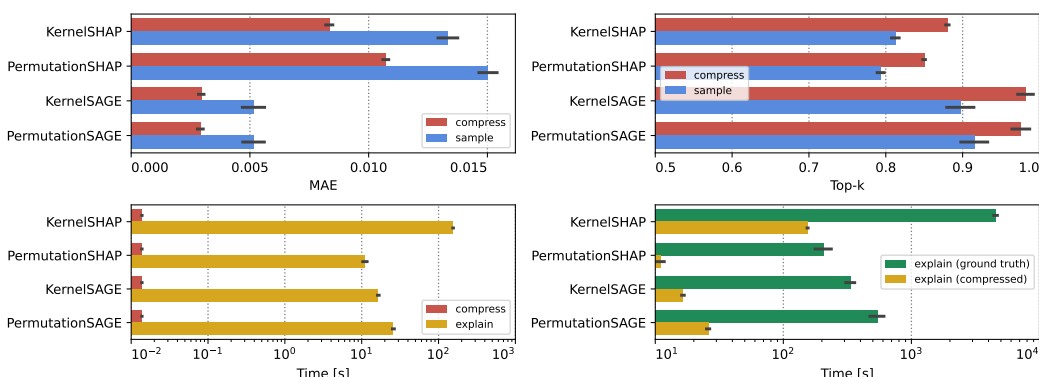

**Figure 12:** Extended Figure 2 (4/4). CTE improves SHAP and SAGE estimation on the gaussian dataset.

## F  CTE IMPROVES GRADIENT-BASED EXPLANATIONS

Figure 13 shows the EXPECTED-GRADIENTS approximation error for 18 datasets. In all cases, CTE achieves on-par approximation error using fewer samples than i.i.d. sampling, i.e. requiring fewer model inferences, resulting in faster computation and saved resources.

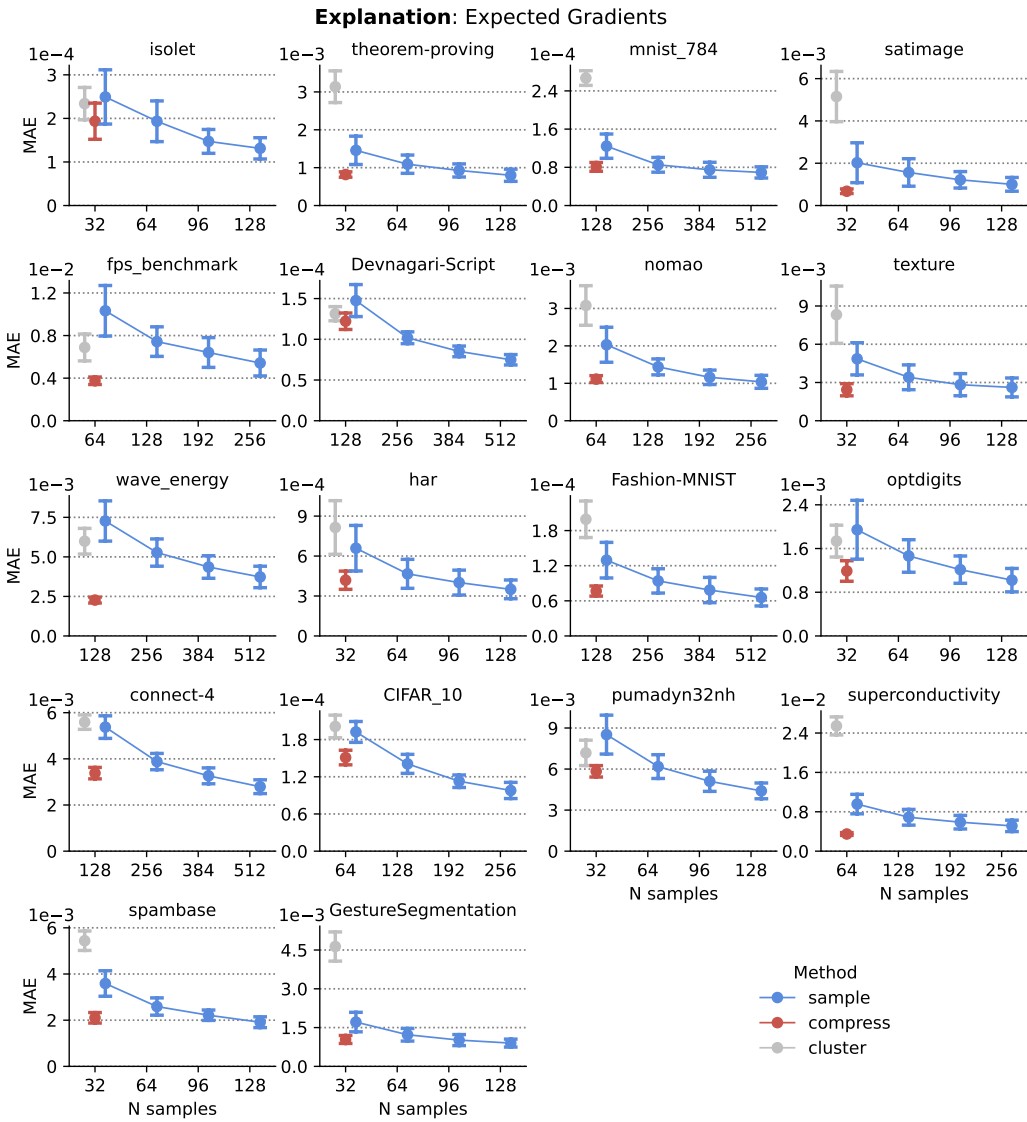

**Figure 13:** Extended Figure 5. Comparison between CTE, i.i.d. sampling and clustering for EXPECTED-GRADIENTS explanations on 18 datasets. We measure mean absolute error (MAE ↓) between feature attribution values. CTE is not only more efficient and accurate, but also more stable as measured with deviation. (mean ± sd.)

**Model-agnostic explanation of a language model.** We further experimented with applying CTE to improve the estimation of global aggregated LIME (Ribeiro et al., 2016), aka G-LIME (Li et al., 2023), which is a more complex setup that we leave for future work. We aim to explain the predictions of a DistilBERT language model[2] trained on the IMDB dataset[3] for sentiment analysis. We calculate LIME with $k = 10$ for all samples from the validation set using an A100 GPU and aggregate these local explanations into global token importance with a mean of absolute normalized values (Li et al., 2023), which is the "ground truth" explanation. We then compress the set with i.i.d. sampling, CTE, and clustering based on the inputs' text embeddings from the model's last layer (preceding a classifier) that has a dimension of size 768. Figure 14 shows results for explanation approximation error and an exemplary comparison between the explanations relating to Figure 1. To obtain these results, we used $8\times$ more samples than the typical compression scenario (still $25\times$ fewer than the full sample) so as to overcome the issue of rare tokens skewing the results. It becomes challenging to compute the distance between the ground truth and approximated explanations as the latter contains significantly fewer tokens (features), as opposed to previous experiments where these two explanations always had equal dimensions. Thus, MAE becomes biased towards sparse explanations and popular tokens, i.e. an explanation with a single token of well-approximated importance could have an error close to 0. For context, we measure TV between the discrete distributions of tokens in local explanations before the global aggregation (lower is better). We report results for different token cutoffs, where we remove the tokens from the ground truth explanation by their rarity, which saturates at 5% tokens left.

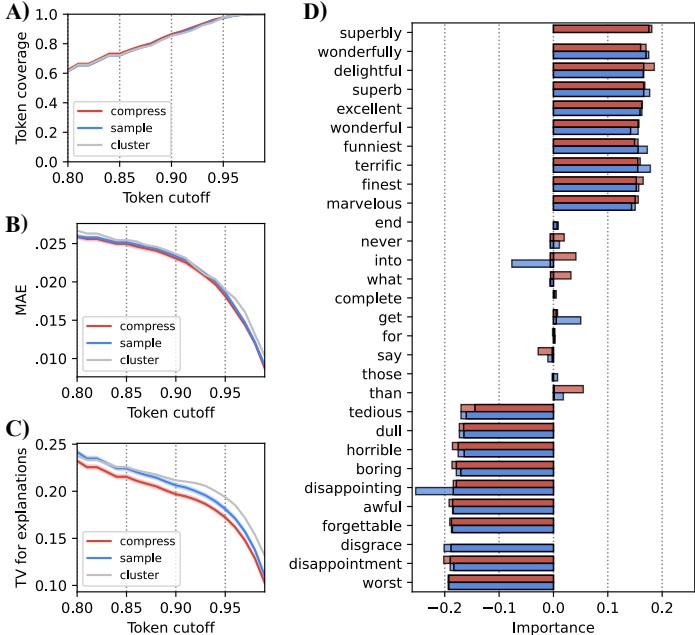

**Figure 14:** CTE for G-LIME of a DistilBERT model classifying IMDB reviews. **A)** It is not obvious how to measure the distance between global explanations containing different sets of tokens (Token coverage in % w.r.t. ground truth, ↑). Therefore, we gradually remove rare tokens from the measurement based on their occurrence in the ground truth explanation (Token cutoff in quantiles). **B)** Measurement of mean absolute error (MAE, ↓) between aggregated global explanations. **C)** Measurement of total variation distance (TV, ↓) between token occurrences in local explanations before global aggregation. **D)** We show an exemplary "worst-case" explanation, i.e. with the lowest MAE for cutoff 0.95 where token coverage is over 99%, for both CTE and i.i.d. sampling. For this visualization, we only show the importance of the 5 most positive/negative tokens, and 5 tokens with the importance closest to zero. Explanation approximation error is indicated with transparent bars. Notably, i.i.d. sampling misses containing any input with an important token "superbly", while CTE misses "disgrace". Sampling overestimates the global importance of tokens "disappointing", "into" and "get", while CTE, for example, overestimates "than" and underestimates "tedious" or "delightful".

---

[2]https://huggingface.co/dfurman/distilbert-base-uncased-imdb
[3]https://huggingface.co/datasets/stanfordnlp/imdb

## G  ABLATIONS

Figures 15–23 report the explanation approximation error for 30 predictive tasks. We observe that CTE significantly improves the estimation of FEATURE-EFFECTS in all cases. Another insight is that, on average, CTE provides a smaller improvement over i.i.d. sampling when considering compressing foreground data in SAGE.

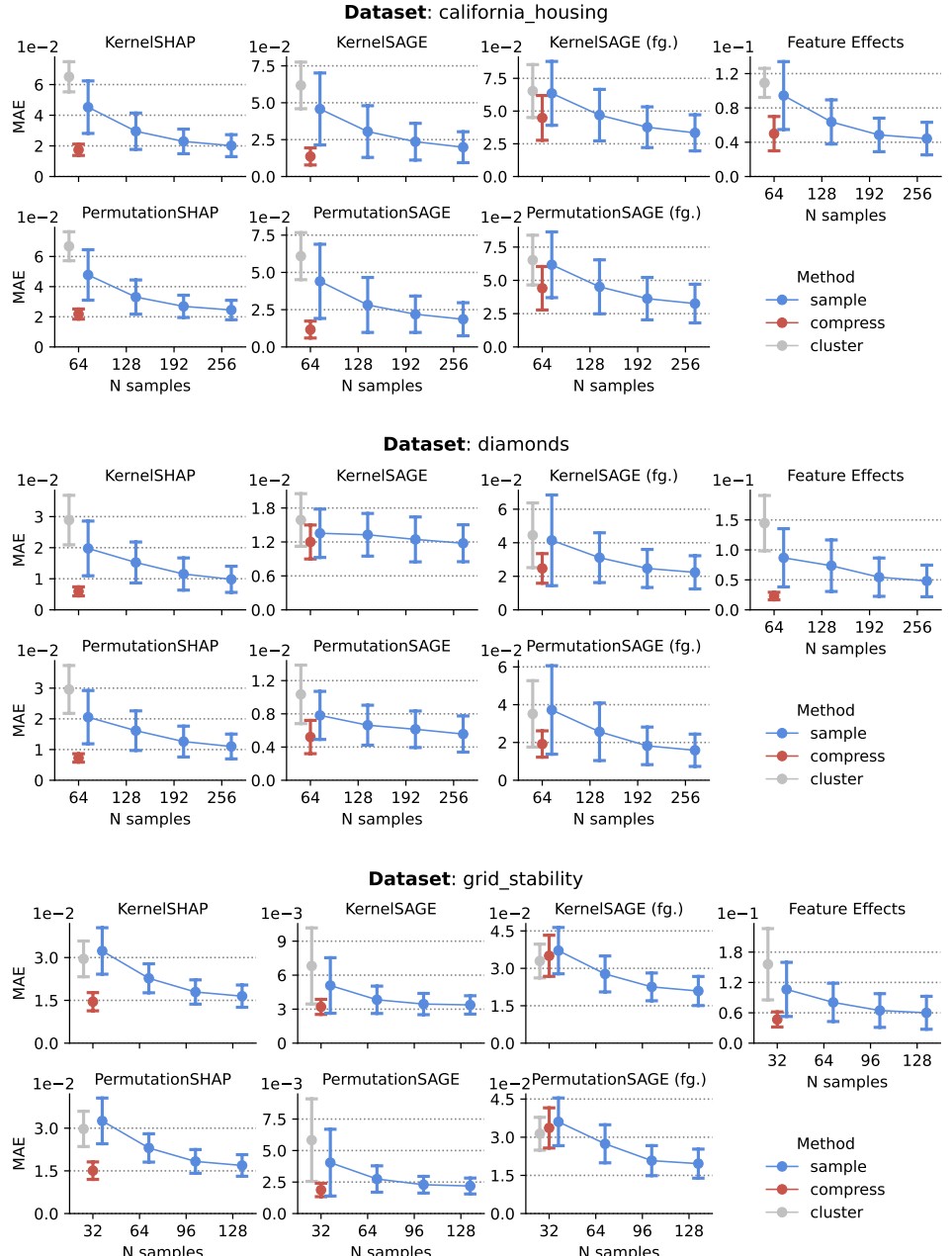

**Figure 15:** Extended Figure 6 (1/10). CTE improves the explanation approximation error of various local and global removal-based explanations. SAGE is evaluated in two variants that consider either compressing only the background data (default), or using the compressed samples as both background and foreground data (as indicated with "fg."). (mean ± sd.)

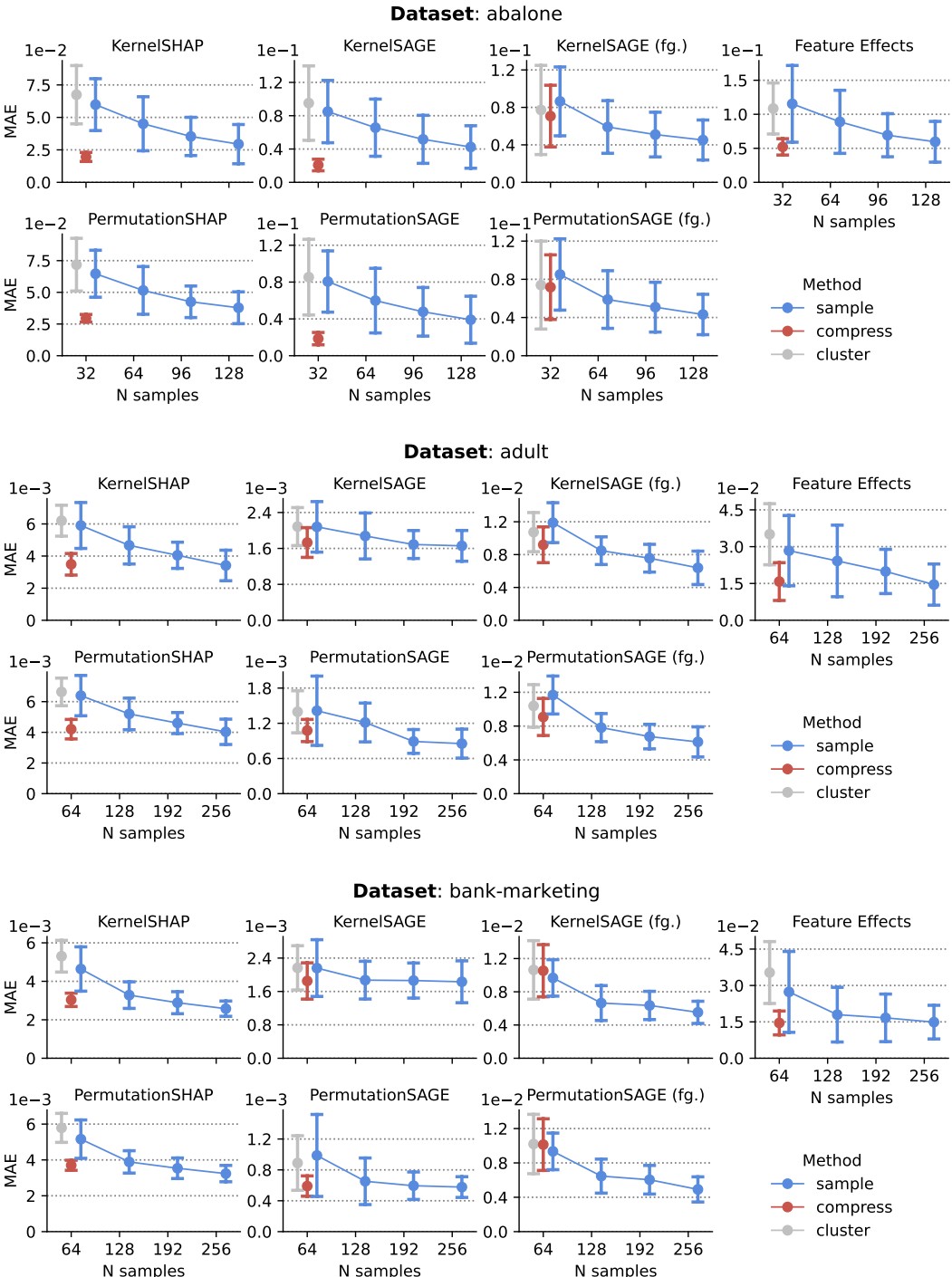

**Figure 16:** Extended Figure 6 (2/10). CTE improves the explanation approximation error of various local and global removal-based explanations. SAGE is evaluated in two variants that consider either compressing only the background data (default), or using the compressed samples as both background and foreground data (as indicated with "fg."). (mean ± sd.)

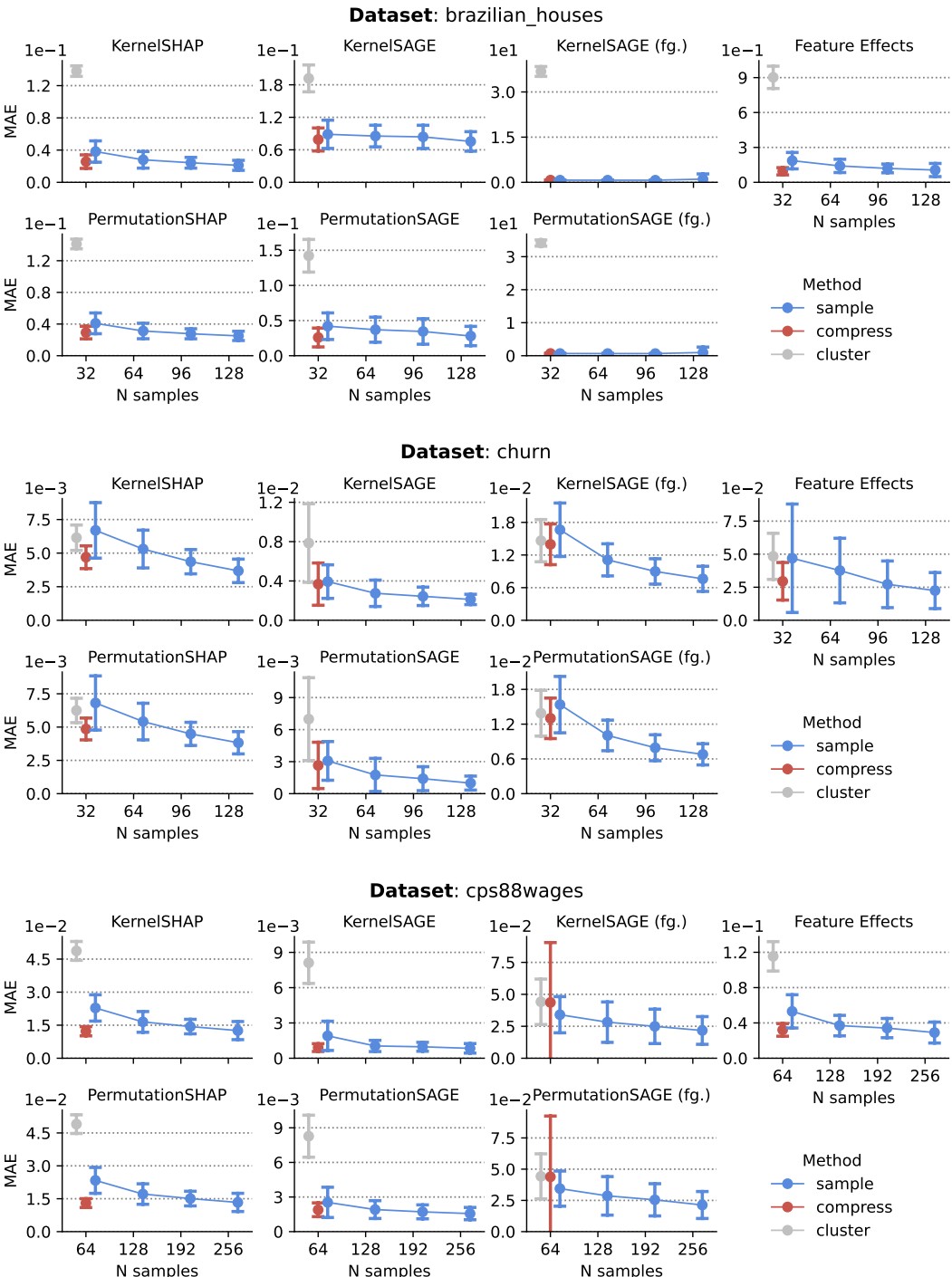

**Figure 17:** Extended Figure 6 (3/10). CTE improves the explanation approximation error of various local and global removal-based explanations. SAGE is evaluated in two variants that consider either compressing only the background data (default), or using the compressed samples as both background and foreground data (as indicated with "fg."). (mean ± sd.)

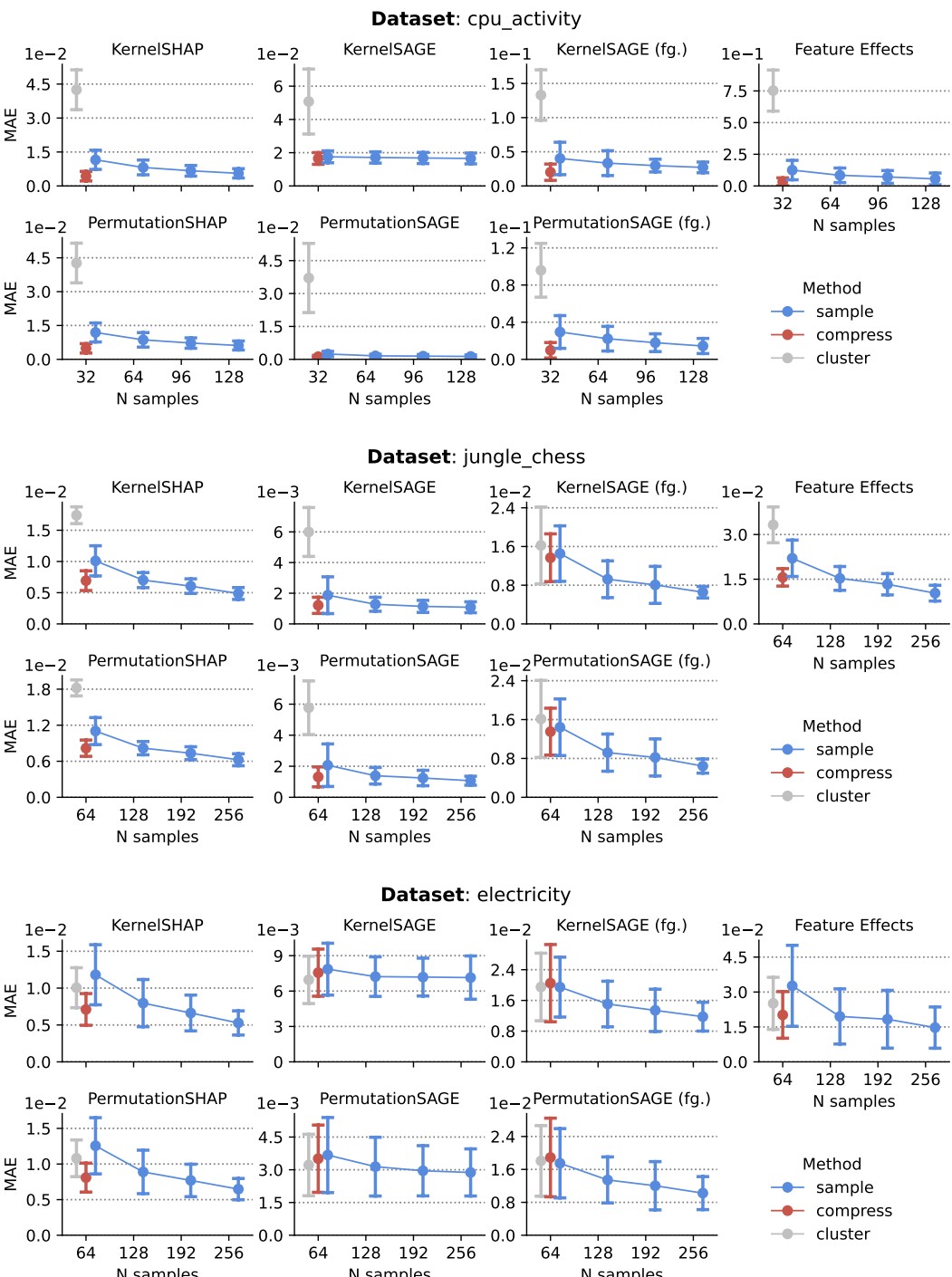

**Figure 18:** Extended Figure 6 (4/10). CTE improves the explanation approximation error of various local and global removal-based explanations. SAGE is evaluated in two variants that consider either compressing only the background data (default), or using the compressed samples as both background and foreground data (as indicated with "fg."). (mean ± sd.)

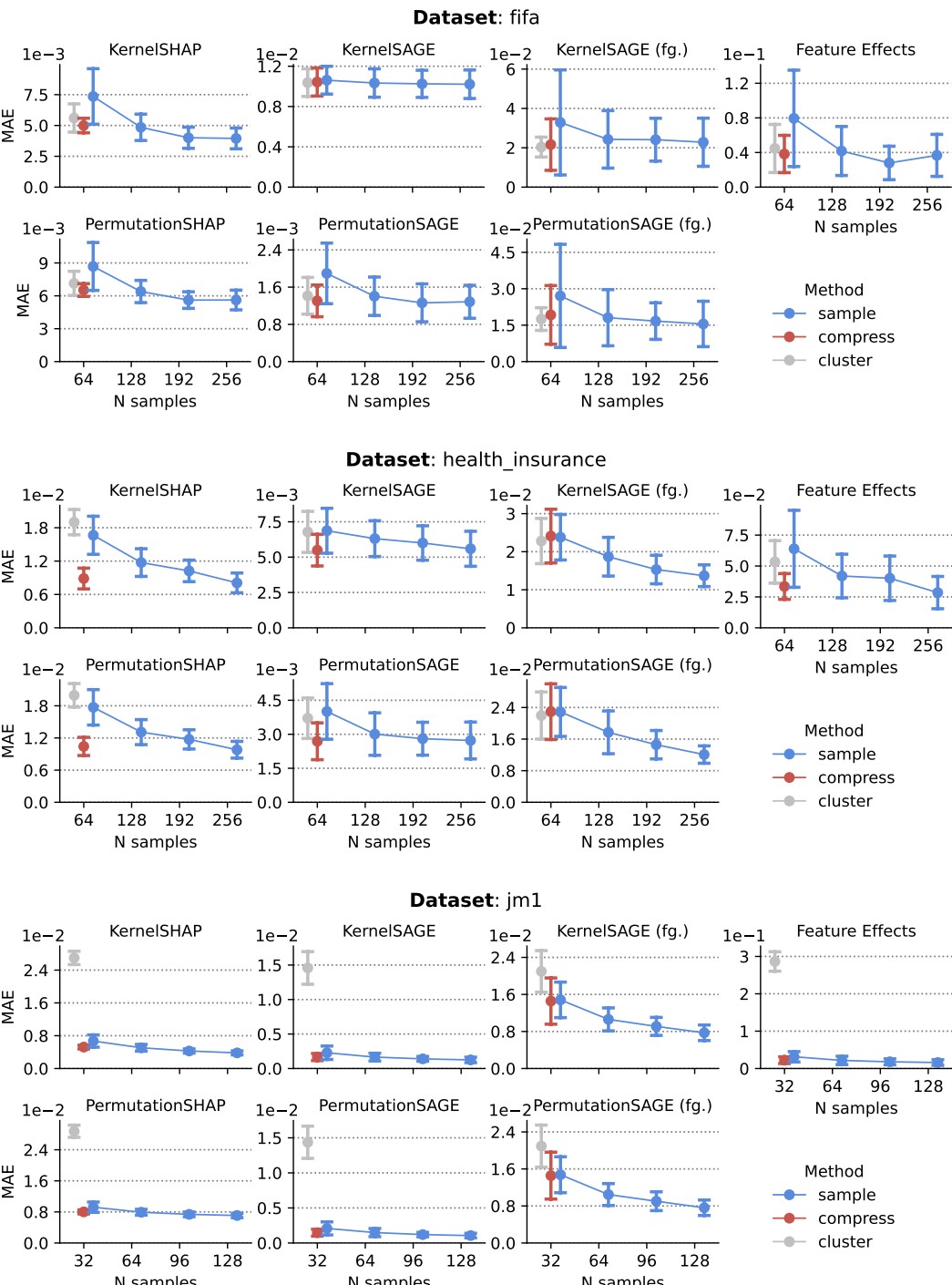

**Figure 19:** Extended Figure 6 (5/10). CTE improves the explanation approximation error of various local and global removal-based explanations. SAGE is evaluated in two variants that consider either compressing only the background data (default), or using the compressed samples as both background and foreground data (as indicated with "fg."). (mean ± sd.)

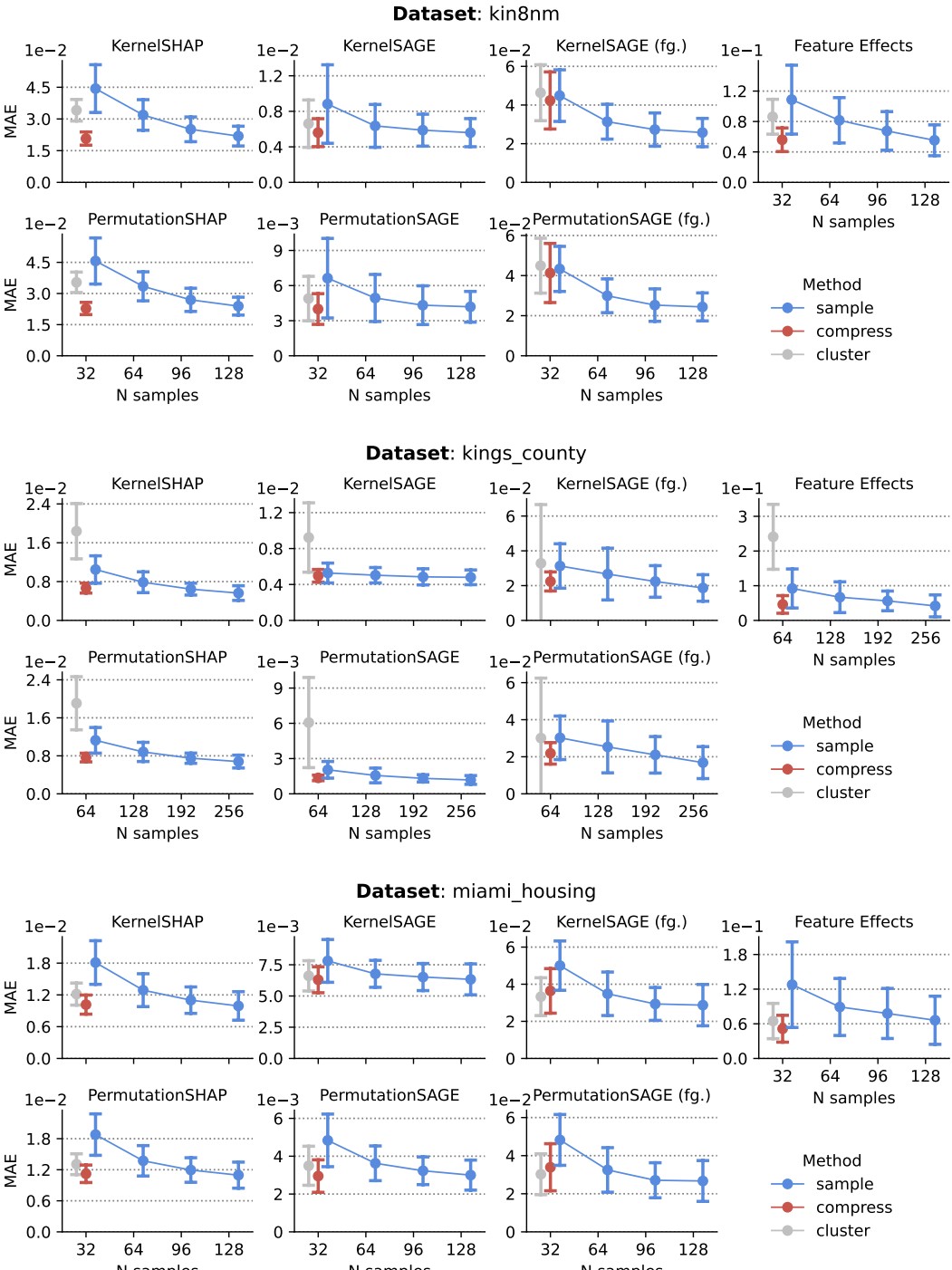

**Figure 20:** Extended Figure 6 (6/10). CTE improves the explanation approximation error of various local and global removal-based explanations. SAGE is evaluated in two variants that consider either compressing only the background data (default), or using the compressed samples as both background and foreground data (as indicated with "fg."). (mean $\pm$ sd.)

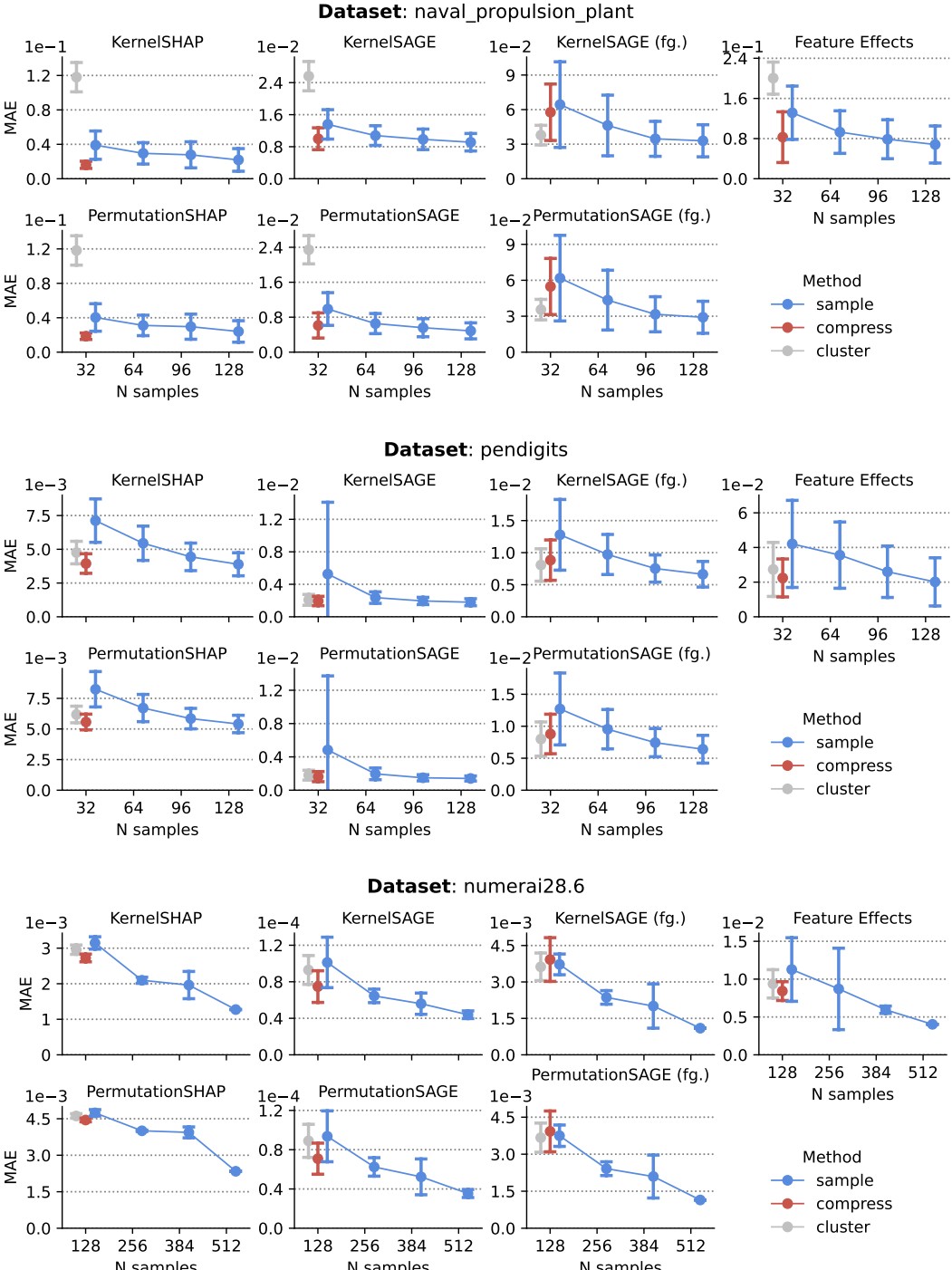

**Figure 21:** Extended Figure 6 (7/10). CTE improves the explanation approximation error of various local and global removal-based explanations. SAGE is evaluated in two variants that consider either compressing only the background data (default), or using the compressed samples as both background and foreground data (as indicated with "fg."). (mean ± sd.)

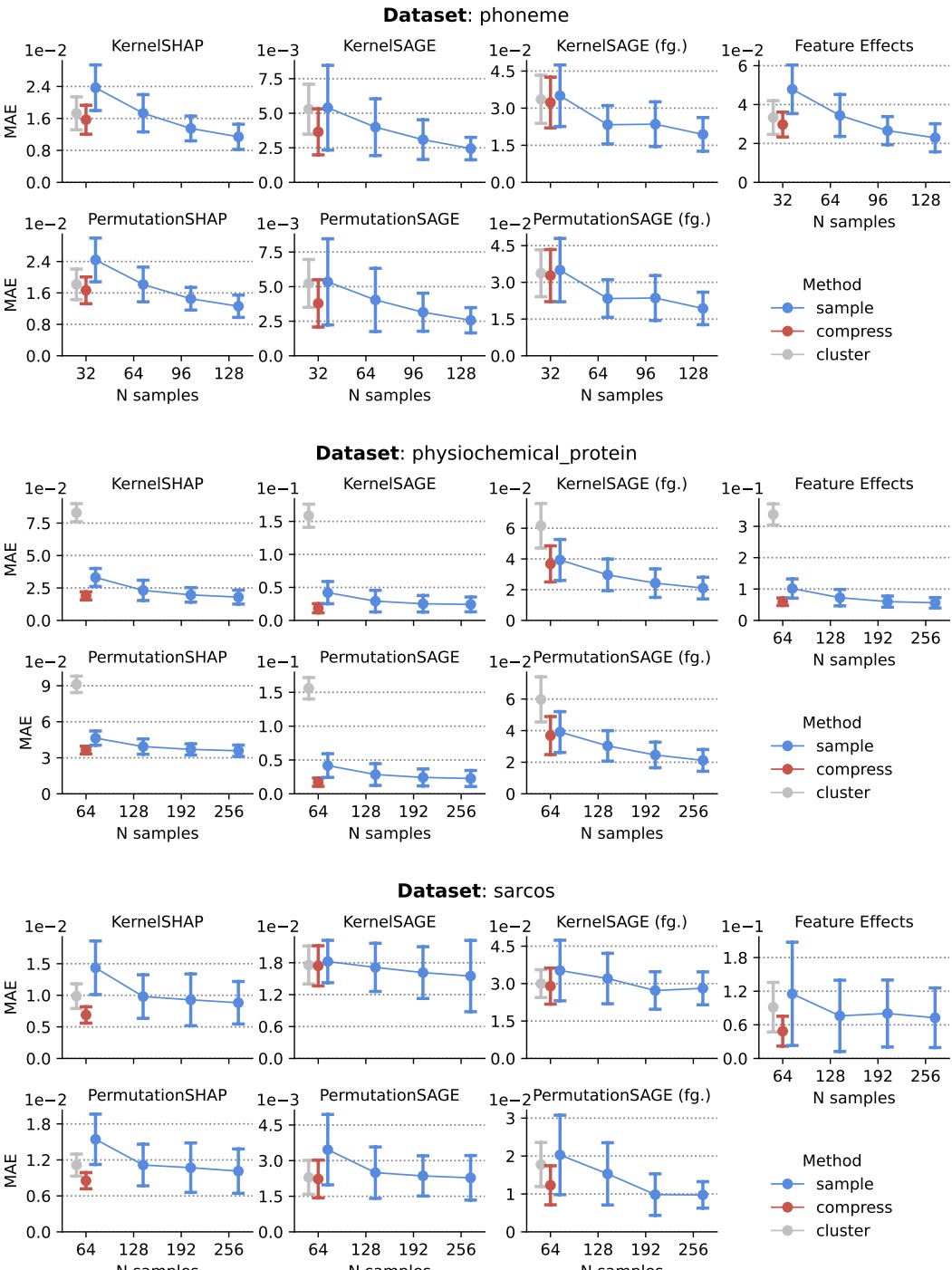

**Figure 22:** Extended Figure 6 (8/10). CTE improves the explanation approximation error of various local and global removal-based explanations. SAGE is evaluated in two variants that consider either compressing only the background data (default), or using the compressed samples as both background and foreground data (as indicated with "fg."). (mean ± sd.)

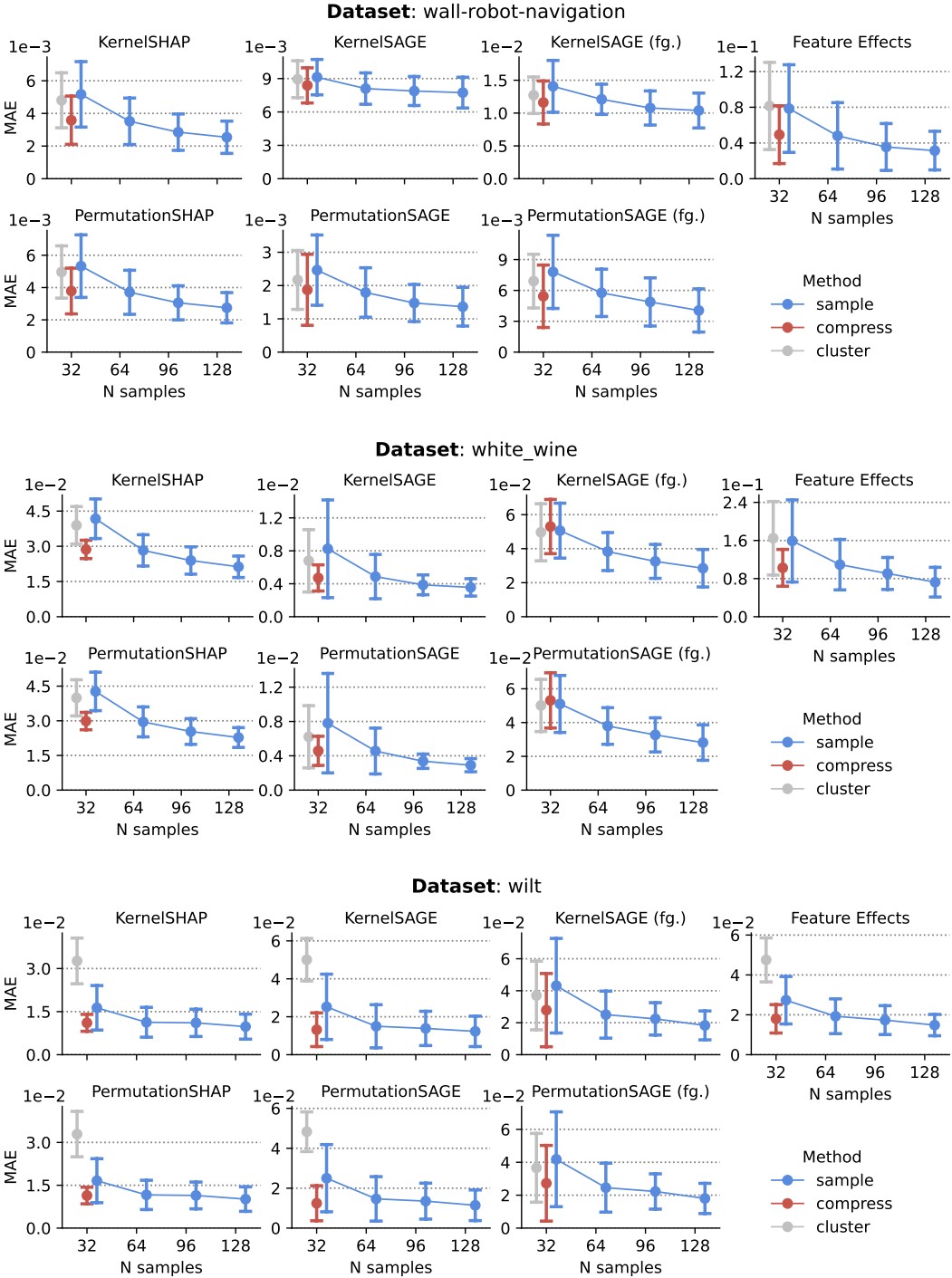

**Figure 23:** Extended Figure 6 (9/10). CTE improves the explanation approximation error of various local and global removal-based explanations. SAGE is evaluated in two variants that consider either compressing only the background data (default), or using the compressed samples as both background and foreground data (as indicated with "fg."). (mean ± sd.)

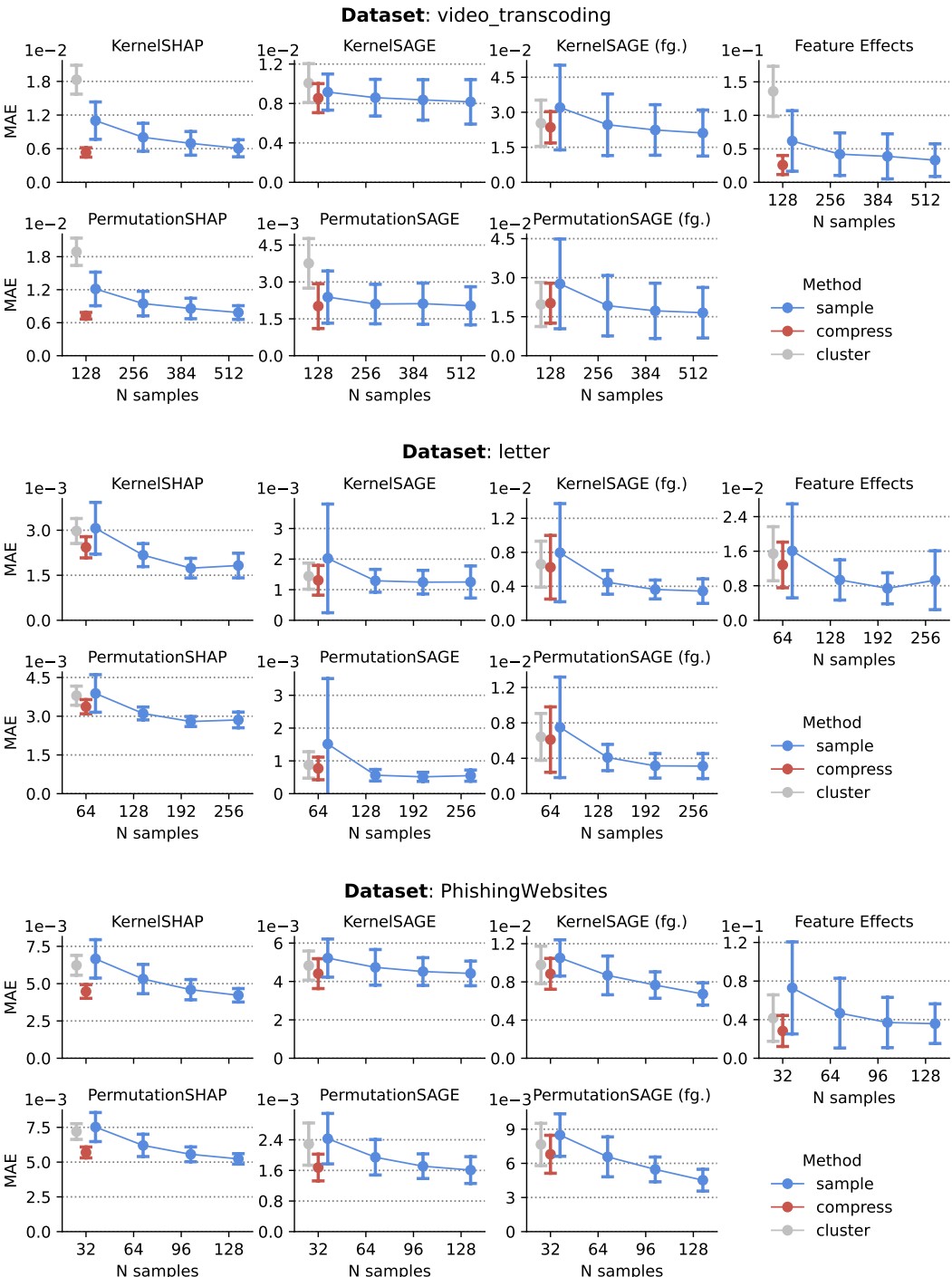

**Figure 24:** Extended Figure 6 (10/10). CTE improves the explanation approximation error of various local and global removal-based explanations. SAGE is evaluated in two variants that consider either compressing only the background data (default), or using the compressed samples as both background and foreground data (as indicated with "fg."). (mean ± sd.)

## H  COMPUTE RESOURCES

Experiments described in Sections 4.1, 4.2 & 4.5, and Figure 4, were computed on a personal computer with an M3 chip as justified in the beginning of Section 4. Experiments described in Sections 4.3 & 4.4 were computed on a cluster with $4\times$ AMD Rome 7742 CPUs (256 cores) and 4TB of RAM for about 14 days combined.

## I VISUAL COMPARISON OF EXPLANATIONS

We provide exemplary visual comparisons between ground truth explanations and those estimated on an i.i.d. and compressed sample in 4 experimental settings.

Figure 25 shows a comparison for KERNEL-SAGE explaining an XGBoost model trained on the `compas` dataset. Figure 26 shows a comparison for PERMUTATION-SHAP explaining a neural network trained on the `compas` dataset, and Figure 27 shows the same on the `heloc` dataset. Note that we show all local explanations at once (not to hand-pick a single one), which might falsely look like a good approximation "on average" when in fact the attribution values in specific cases differ significantly. Figure 28 shows a comparison for FEATURE-EFFECTS explaining an XGBoost model trained on the `grid_stability` dataset, and Figure 29 shows the same on the `miami_housing` dataset. Figures 30 & 31 show exemplary visual comparisons for EXPECTED-GRADIENTS explaining a convolutional neural network trained on the `CIFAR_10` dataset.

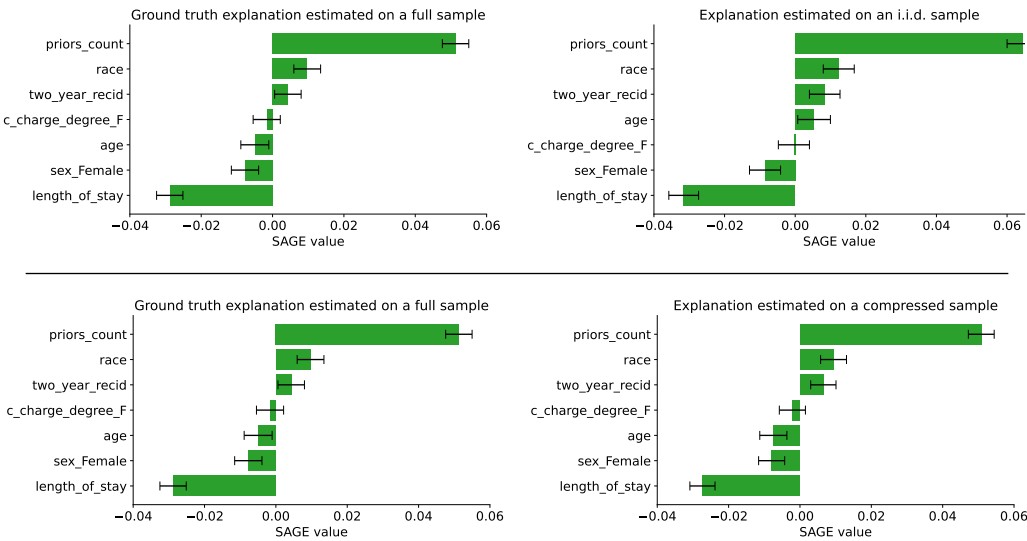

**Figure 25:** Comparison between KERNEL-SAGE estimated on the full (left), sampled (right top), and compressed (right bottom) subsets of the `compas` dataset. MAE introduced by i.i.d. sampling equals 0.0050 for the importance values and 0.00033 for their standard deviations (error bars), by CTE is 0.0011 and 0.00007 respectively, and so the relative improvement of CTE is 78% in both cases.

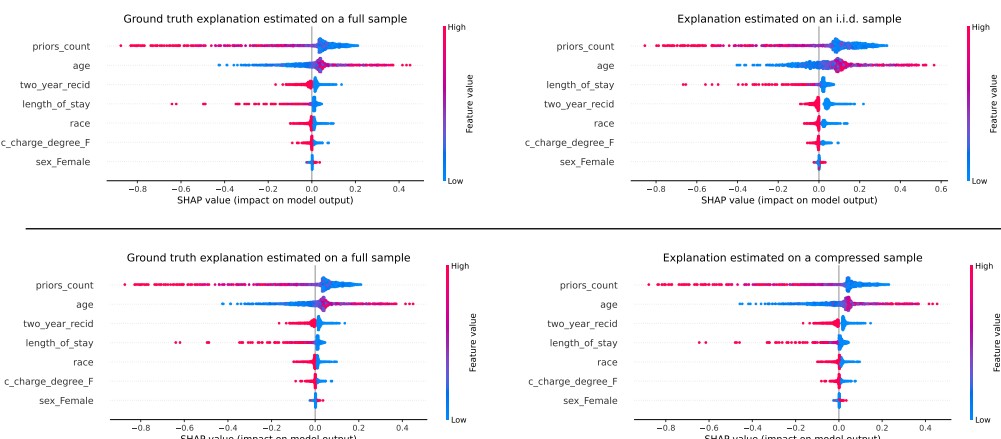

**Figure 26:** Comparison between all local PERMUTATION-SHAP explanations estimated on full (left), sampled (right top), and compressed (right bottom) subsets of the `compas` dataset. MAE introduced by i.i.d. sampling equals 0.0227, by CTE is 0.0032, and so the relative improvement of CTE is 86%.

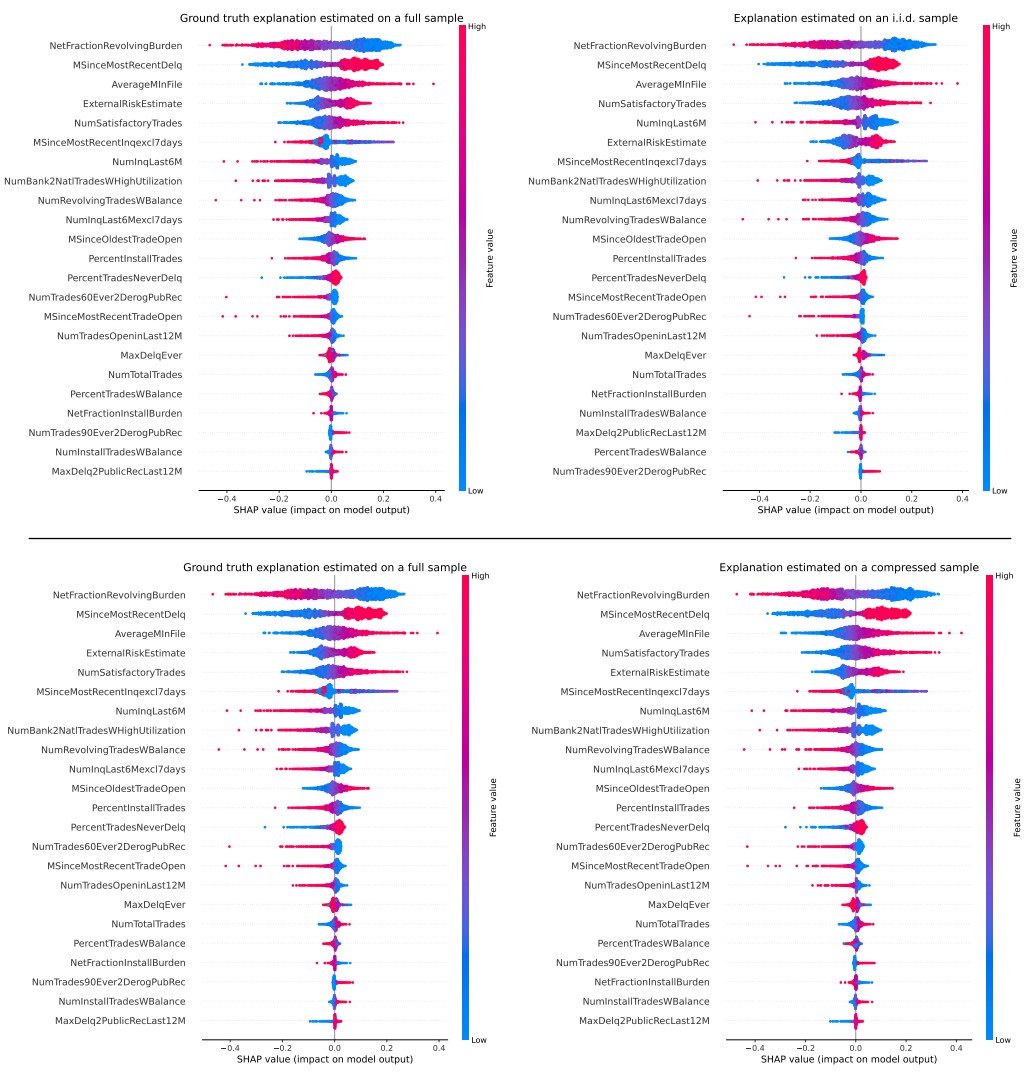

**Figure 27:** Comparison between all local PERMUTATION-SHAP explanations estimated on full (left), sampled (right top), and compressed (right bottom) subsets of the `heloc` dataset. MAE introduced by i.i.d. sampling equals 0.0087, by CTE is 0.0053, and so the relative improvement of CTE is 38%.

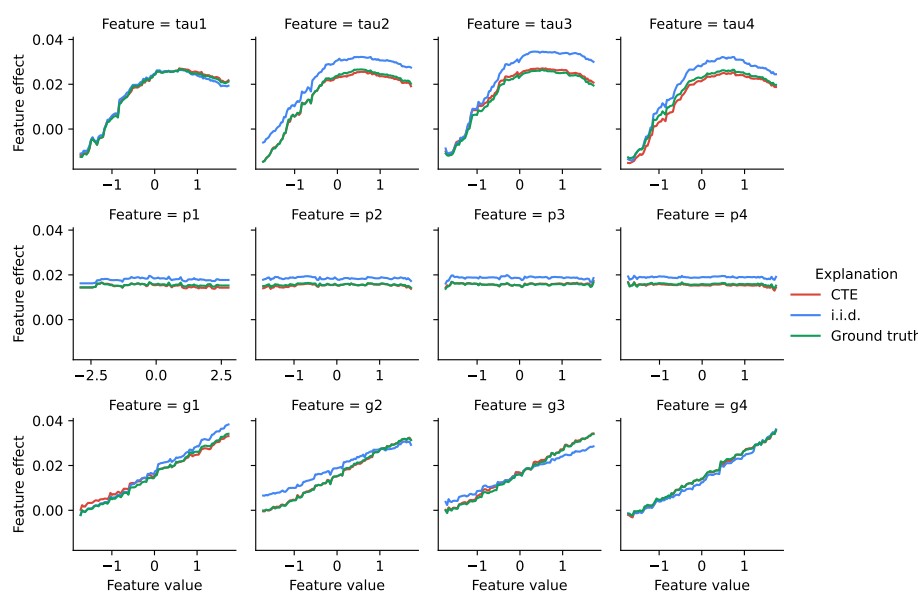

**Figure 28:** Comparison between FEATURE-EFFECTS explanation estimated on the full (Ground truth), sampled (i.i.d.), and compressed (CTE) subsets of the `grid_stability` dataset. MAE introduced by i.i.d. sampling equals 0.0032, by CTE is 0.0007, and so the relative improvement of CTE is 79%.

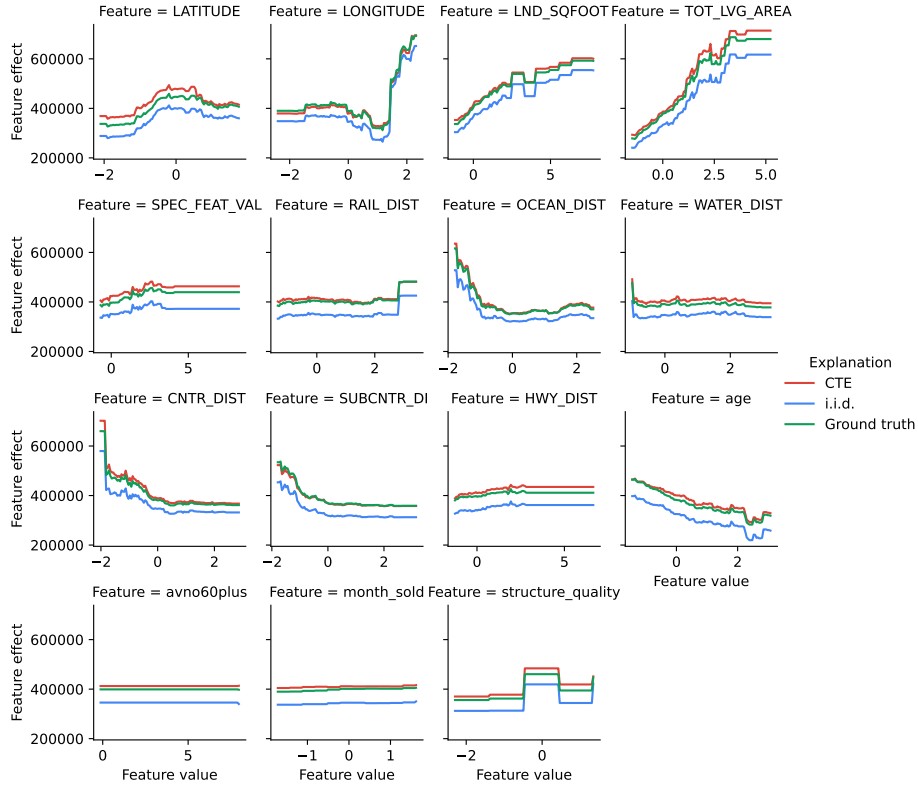

**Figure 29:** Comparison between FEATURE-EFFECTS explanation estimated on the full (Ground truth), sampled (i.i.d.), and compressed (CTE) subsets of the `miami_housing` dataset. MAE introduced by i.i.d. sampling equals 49766, by CTE is 14031, and so the relative improvement of CTE is 71%.

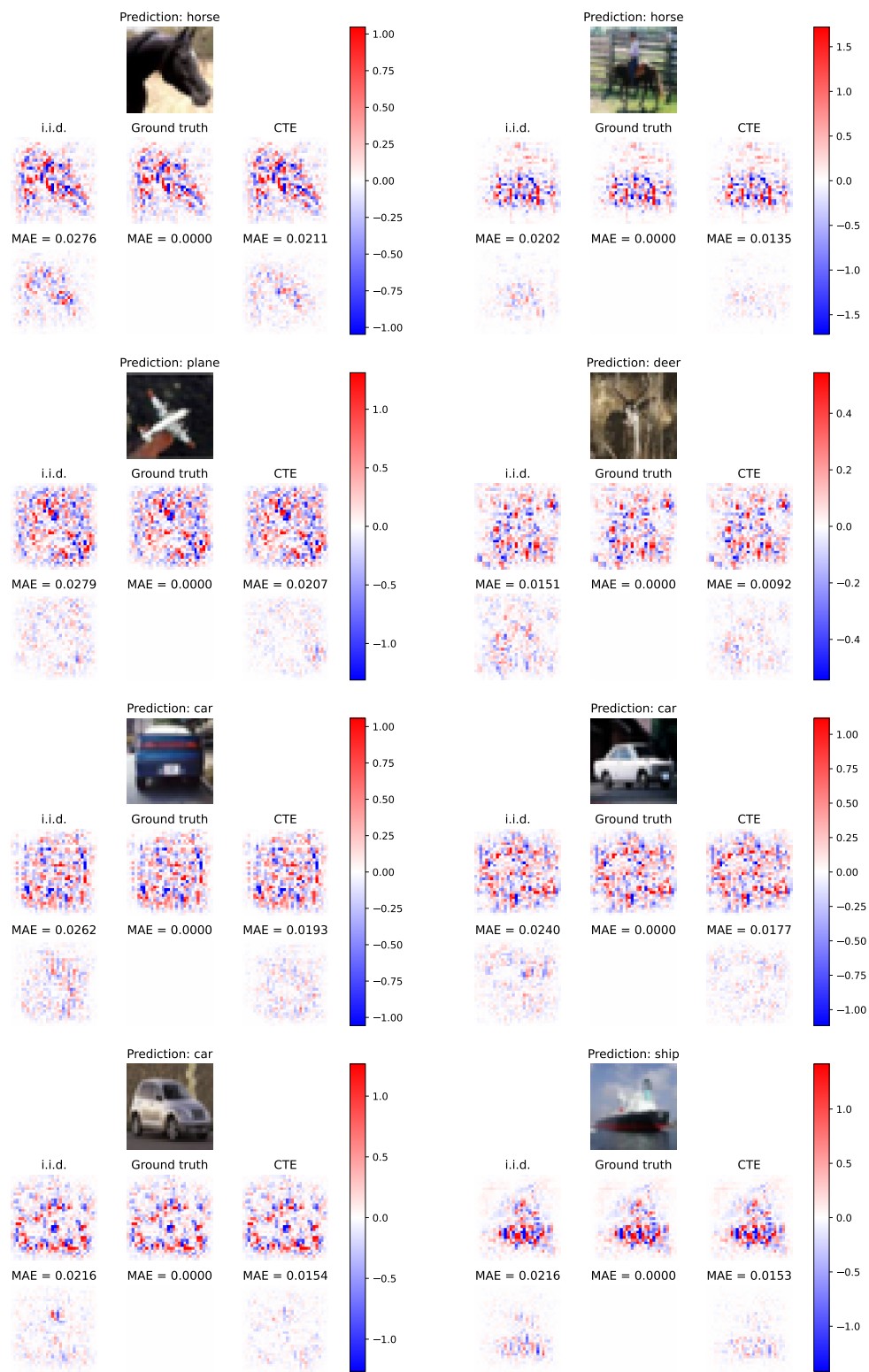

**Figure 30:** Comparison between EXPECTED-GRADIENTS explanations estimated on the full (Ground truth), sampled (i.i.d.), and compressed (CTE) subsets of the CIFAR₋10 dataset. The bottom rows visualize the differences (MAE ↓) from the ground truth explanation. All predictions are correct.

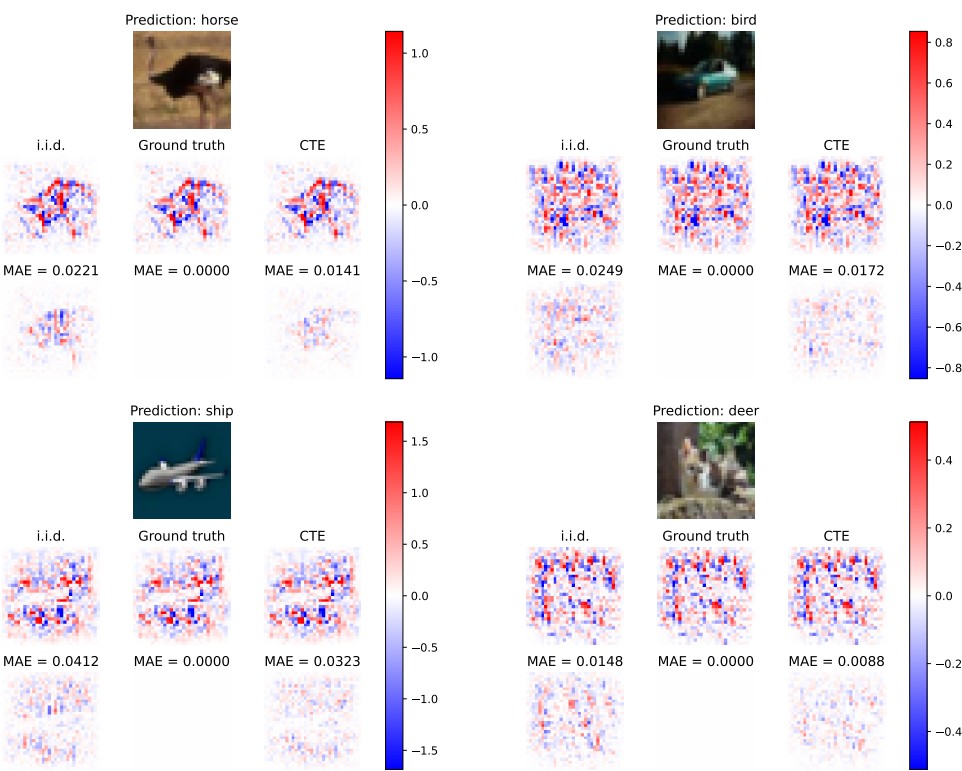

**Figure 31:** Comparison between EXPECTED-GRADIENTS explanations estimated on the full (Ground truth), sampled (i.i.d.), and compressed (CTE) subsets of the CIFAR_10 dataset. The bottom rows visualize the differences (MAE ↓) from the ground truth explanation. All predictions are wrong.

