# OpenReview forum: "Efficient and Accurate Explanation Estimation with Distribution Compression"
_ICLR.cc/2025/Conference — ICLR 2025 Spotlight_

### Official Review · Reviewer_dB4i · 2024-10-23

**Soundness:** 3
**Presentation:** 3
**Contribution:** 3
**Rating:** 8
**Confidence:** 3

**Summary:**

This paper tackles the problem of efficiently approximating the marginal distribution that is often used in post hoc explanation methods. The authors propose using kernel thinning to obtain a compressed sample of the "background" samples that are used to empirically estimate the marginalization. The proposed compress then explain method is then empirically demonstrated to require orders of mangitude fewer samples to achieve on-par explanation approximation error compared to i.i.d sampling which is generally utilized.

**Strengths:**

The paper is well written. The biggest strength of the paper is the thoroughness of the experiments. For a paper that is largely an empirical demonstration it is great to see the rigour that clearly went in to the experiment section. The proposal is a neat application of an existing technique in a new domain.

**Weaknesses:**

While the experiment section is rigorously presented, there's room for improvement in presentation - specifically the figures and figure captions. Most of the figures have multiple plots and a single caption. It'd be better to label the sub-plots in each figure as a, b, c,... and then use the caption to describe each figure succinctly. This allows the reader to look at the figures and understand them, rather than having to jump back and forth between the main text and figures.

The other weakness is the fact that this is largely an application of an existing method for efficient sampling. It *is* still a useful contribution to apply this method to explanations, this could be an even better publication if the sampling method was also improved specifically for post explanation methods.

**Questions:**

I'd be interested in hearing the authors' thoughts on the second "weakness" above.

---

> ### Author Response · Authors · 2024-11-15
> **Response by Authors**
>
> **We sincerely thank the anonymous reviewer for their appreciation of our work and the feedback.**
>
> > The biggest strength of the paper is the thoroughness of the experiments. For a paper that is largely an empirical demonstration it is great to see the rigour that clearly went in to the experiment section.
>
> Thank you for highlighting the **thoroughness and rigour of our experiments**.
>
> > **W1:** While the experiment section is rigorously presented, there's room for improvement in presentation - specifically the figures and figure captions. Most of the figures have multiple plots and a single caption. It'd be better to label the sub-plots in each figure as a, b, c,... and then use the caption to describe each figure succinctly. This allows the reader to look at the figures and understand them, rather than having to jump back and forth between the main text and figures.
>
> **A1:** Thank you for this valuable suggestion to improve the presentation of our figures. As indicated, we now labeled sub-plots as "A), B), C), …" and extended the captions in **Figures 2, 4, 6 & 8** (denoted with the green text).
>
> > **W2:** The other weakness is the fact that this is largely an application of an existing method for efficient sampling. It is still a useful contribution to apply this method to explanations, this could be an even better publication if the sampling method was also improved specifically for post explanation methods. [I'd be interested in hearing the authors' thoughts on the second "weakness" above.]
>
> **A2:** We agree with the reviewer that improving the sampling method would further strengthen the publication. Here are our thoughts:
>
> 1. Our paper formulates a **new research problem** of "sample-efficient explainability" and proposes a **robust solution** for it that reduces the error and variance across various experimental settings.
> 2. It was already challenging to discover the appropriate method to significantly improve the approximation of **multiple explanation methods simultaneously**, including feature attributions, importance, and effects.
> 3. We find a valuable **theoretical connection** between the (previously) unrelated domains of explanation estimation and distribution compression.
> 4. In some scenarios, a case exists for applying an existing method with known properties and limitations. Please note that distribution compression algorithms are relatively new to supervised learning and can already be considered sophisticated by many. **The solid theoretical foundations of distribution compression will benefit users of our proposed CTE framework and future research to improve it.**
>
> We share the reviewer's excitement for future work on sample-efficient explainability that will bring more sophisticated sampling methods to improve post-hoc explanations further.

---

> > ### Comment · Reviewer_dB4i · 2024-11-26
> > **Thank you for the response**
> >
> > I'm happy with the author's responses to my and other reviewer's comments and will stay with my original rating.

---

### Official Review · Reviewer_ry7y · 2024-11-01

**Soundness:** 3
**Presentation:** 3
**Contribution:** 3
**Rating:** 8
**Confidence:** 2

**Summary:**

The paper proposes distribution compression as an alternative to iid sampling for explanation estimation. The proposed approach is supported by theoretical guarantees. It also improves computation cost while still achieving errors comparable with iid sample empirically.

**Strengths:**

- The proposed method is solid and supported by theoretical bounds on the error.

- This topic is outside my expertise but I was able to follow the paper. So I think the authors did a good job summarizing the related work and providing relevant background information.

**Weaknesses:**

- I overall enjoyed reading this paper.

- I am not an expert on this area. But it looks like going from iid sampling (the paper's main comparison point) to this more complicated CTE framework (based on kernel thinning) is a big step. I am wondering if there are any other baselines in between these two extremes. For instance, a sampling strategy but not exactly uniform? If there are such baselines, I think it would be good to see a comparison against them too to understand the tradeoff between how complicated the compression scheme is vs how much we can reduce the data size vs the error.

**Questions:**

-

---

> ### Author Response · Authors · 2024-11-15
> **Response by Authors to Reviewer ry7y**
>
> > The proposed method is solid and supported by theoretical bounds on the error.
>
> Thank you for highlighting **our theoretical contribution**.
>
> > This topic is outside my expertise but I was able to follow the paper. So I think the authors did a good job summarizing the related work and providing relevant background information.
>
> Thank you very much for these kind words.
>
> > **W1:** I overall enjoyed reading this paper.
>
> **A1:** We are glad.
>
> > **W2:** I am not an expert on this area. But it looks like going from iid sampling (the paper's main comparison point) to this more complicated CTE framework (based on kernel thinning) is a big step. I am wondering if there are any other baselines in between these two extremes. For instance, a sampling strategy but not exactly uniform? If there are such baselines, I think it would be good to see a comparison against them too to understand the tradeoff between how complicated the compression scheme is vs how much we can reduce the data size vs the error.
>
> **A2:** This is a great question. Since we initiate a new problem formulation, another alternative baseline between these two extremes is **a clustering approach**, which we compare against. We compare CTE to **k-medoids clustering** in experiments (see **Figures 5–7**) and show that our CTE framework outperforms it in explanation estimation. **Figure 8** shows that clustering provides a worse approximation of data distribution, presumably because it provides **no theoretical guarantees**, unlike our approach (**Propositions 1 & 2**). While it is always possible to consider more baselines, we included methods that were either "standard" in this regime or "immediately came to mind as an option".

---

> > ### Comment · Reviewer_fa33 · 2024-11-26
> >
> > Thank you for the authors to apply the recommendations on the paper. I have reviewed their response and here is my evaluation from their rebuttal.
> >
> >
> > One of the notable point that authors mentioned is the **'ground truth' explanation** . However, the notion of **'ground truth explanations'** remains unclear. While the reviewers referenced datasets from papers [2, 3, 4], they did not provide concrete examples of ground truth explanations. To address this concern, the authors could include several examples from the dataset, showcasing the ground truth explanations alongside the explanations generated by their approach, with comparisons to the baseline (e.g., iid).
> >
> >
> > Also, I have asked about the the visual representation of generated explanation. As a paper which has the title of "Efficient and Accurate **Explanation Estimation** with Distribution Compression" it is necessary to provide qualitative comparison on the final explanation as it has been done on the previous works [5-7]. Based on the authors claim, they only provided one example for IMDB dataset which only demonstrate the assigned importance to each token, but the visual presentation of original data, the assigned class label are missing. Also, the authors provided the MAE for MNIST, ICLR, Fashion MNIST in Figure 5, but they didn't provide any qualitative representation of generated explanation.
> >
> > In explanation-focused research, it is standard practice to include qualitative comparisons to substantiate claims of significant improvements over prior methods. Such comparisons should be included in the main paper rather than relegated to the Appendix.
> >
> > in Figure 5, the improvements reported in MAE for the proposed method appear to be in the order of 1×10−31×10−3 to 1×10−41×10−4. Could the authors elaborate on these results and clarify why the performance improvement is minimal for datasets like CIFAR-10 and MNIST?
> >
> >
> > Regarding the Kernel comparison, in the original paper [1] the authors have shown that how change of kernel significantly influences on selected samples which can later influence the proposed method. Investigation of different kernel could be an a great discussion on the paper and how it influences the quality of explanation.
> >
> >
> > My intuition for comparison with Influence Function was selecting the top influential sample for a given test data and how these influential samples differ from the top explanation that your method obtains.
> >
> > [1] Dwivedi, Raaz, and Lester Mackey. "Generalized kernel thinning." _arXiv preprint arXiv:2110.01593_ (2021).
> >
> > [2] Agarwal, Chirag, et al. "Openxai: Towards a transparent evaluation of model explanations." _Advances in neural information processing systems_ 35 (2022): 15784-15799.
> >
> > [3] Bischl, Bernd, et al. "Openml benchmarking suites." _arXiv preprint arXiv:1708.03731_ (2017).
> >
> > [4] Fischer, Sebastian Felix, Matthias Feurer, and Bernd Bischl. "OpenML-CTR23–a curated tabular regression benchmarking suite." _AutoML Conference 2023 (Workshop)_. 2023.
> >
> > [5] Ribeiro, Marco Tulio, Sameer Singh, and Carlos Guestrin. "" Why should i trust you?" Explaining the predictions of any classifier." _Proceedings of the 22nd ACM SIGKDD international conference on knowledge discovery and data mining_. 2016.
> >
> > [6] Lundberg, Scott. "A unified approach to interpreting model predictions." _arXiv preprint arXiv:1705.07874_ (2017).
> >
> > [7] Koh, Pang Wei, and Percy Liang. "Understanding black-box predictions via influence functions." _International conference on machine learning_. PMLR, 2017.

---

> > ### Comment · Reviewer_ry7y · 2024-11-30
> > **Thank you for the clarification**
> >
> > I thank the authors for their response. I still think it is a good paper and recommend acceptance.

---

> ### Author Response · Authors · 2024-11-27
> **Response by Authors to the Official Comment by Reviewer fa33 (not Reviewer ry7y)**
>
> **Dear all, please note that we here respond to the above Response by Reviewer fa33 (and not Reviewer ry7y), which is mistakenly posted under the Review by Reviewer ry7y.**
>
> > One of the notable point that authors mentioned is the 'ground truth' explanation . However, the notion of 'ground truth explanations' remains unclear. While the reviewers referenced datasets from papers [2, 3, 4], they did not provide concrete examples of ground truth explanations. To address this concern, the authors could include several examples from the dataset, showcasing the ground truth explanations alongside the explanations generated by their approach, with comparisons to the baseline (e.g., iid). Also, I have asked about the the visual representation of generated explanation. As a paper which has the title of "Efficient and Accurate Explanation Estimation with Distribution Compression" it is necessary to provide qualitative comparison on the final explanation as it has been done on the previous works [5-7]. Based on the authors claim, they only provided one example for IMDB dataset which only demonstrate the assigned importance to each token, but the visual presentation of original data, the assigned class label are missing. Also, the authors provided the MAE for MNIST, ICLR, Fashion MNIST in Figure 5, but they didn't provide any qualitative representation of generated explanation. In explanation-focused research, it is standard practice to include qualitative comparisons to substantiate claims of significant improvements over prior methods. Such comparisons should be included in the main paper rather than relegated to the Appendix.
>
> Thank you for raising these points. We have now added exemplary visual comparisons between ground truth explanations and those estimated on an i.i.d. and compressed sample in **Appendix I**. We hope this suffices as we see no space to put them in the main paper; also at this moment, after positive feedback from Reviewers dB4i and ry7y.
>
> > in Figure 5, the improvements reported in MAE for the proposed method appear to be in the order of 1×10−31×10−3 to 1×10−41×10−4. Could the authors elaborate on these results and clarify why the performance improvement is minimal for datasets like CIFAR-10 and MNIST?
>
> We have now added a discussion on these results in the paper **lines 397–401**. Please note that the order of magnitude can be small because the error is calculated as an average over thousands of features (pixels); we could have reported an unnormalized (larger) sum instead. Regarding specifically CIFAR-10 and MNIST: The performance improvements are significant, i.e. CTE decreases the estimation error for Expected Gradients by 35% on MNIST (Welch's t-test: p-value < 1e-10) and by 21% on CIFAR-10 (p < 1e-10). Moreover, CTE provides 2x efficiency improvements as measured by the number of samples required for i.i.d. to reach the performance of CTE. We deliver results on 14 more datasets in Figure 13 (Appendix F), where our method improves the estimation even further, e.g. by 39% on the GestureSegmentation dataset (p < 1e-10). Furthermore, the variance of estimation (sd. as denoted with error bars in Figures 5 & 13) decreases on all the datasets, which shows CTE improves not only the accuracy but also the stability of explanation estimation over the current state-of-the-art.
>
> > Regarding the Kernel comparison, in the original paper [1] the authors have shown that how change of kernel significantly influences on selected samples which can later influence the proposed method. Investigation of different kernel could be an a great discussion on the paper and how it influences the quality of explanation.
>
> We agree and maintain our original response that investigating different kernel functions is a viable future work direction (**lines 524–526**).
>
> > My intuition for comparison with Influence Function was selecting the top influential sample for a given test data and how these influential samples differ from the top explanation that your method obtains.
>
> Thank you for the engaging discussion. We have now referenced investigating the use of influence functions in our framework as a potential extension in the paper **lines 529–532**: “One could also investigate how influence functions (Koh & Liang, 2017), which aim to attribute the importance of data to the model’s prediction, can guide sampling for efficient explanation estimation.”

---

### Official Review · Reviewer_fa33 · 2024-11-03

**Soundness:** 3
**Presentation:** 2
**Contribution:** 3
**Rating:** 6
**Confidence:** 5

**Summary:**

the paper introduces CTE uses distribution compression through kernel thinning to obtain a data sample to better and more efficiently approximate the marginal distribution. This method is recommend to significantly reduce the computational complexity of attribution-based methods such as SHAP and SAGE.

**Strengths:**

This approach significantly reduces the computational complexity of SHAP and SAGE.

**Weaknesses:**

line 53 - "We introduce a new paradigm for estimating post-hoc explanations based on a marginal distribution" ---> The main contribution of paper is not clearly described. From the introduction, reader might get the idea that you are proposing a new explanation method that employs KT, but as we go further into the paper, the tone of authors changes and they focus on the effectiveness their approach for Kernel SHAP and SAGE.


the idea of this paper needs to be more clearly stated. From my understanding the paper attempts to improve the computation time of SHAP and SAGE using Kernel thinning. The Kernel Thinning is proposed to better select the samples for the feature attribution explanation. The idea seems interesting and this can help with the Shapely and Sage method, but this approach is only specific to these explanation method that are based on the selection of subsets of data.

The experimental evaluation is heavily focused on MAE with  the explanation as "...". I wonder if this approach is the main criteria that paper investigated, what is the main baseline that they compare with to obtain the explanation error. If the explanation error is calculated with the SHAP with the iid sampling, so what does Table 1 mean. because in this table the paper explains that from the iid sampling the MAE of explain improved to something that is achieved with CTE.


Also the paper only explored the Gaussian Kernel. Why that choice? why not exploring other kernels? What is the theoretical aspect of Gaussian that advantages your approach?

also, I wanted to see some examples of how the trained explainer using your method performs for image and examples of IMDB dataset. The paper heavily focuses on the MAE without providing any example to show the qualitative comparison of generated explanation


The main purpose of KT, is reducing the computational complexity and the quality of generated explanation, but since this method is focused on the data attribution concept, I was expecting see a comparison between the KT and Influence Function in extracted explanations.

**Questions:**

line 184 - definition 4 - it is not clear what is the output f this function and how do you compare it with for calculation of MAE. how the explanation error is defined and how it is calculated? as far as I understand the mean absolute error is measured based on the difference between the assigned values to  features of explanation from the SHAP  with iid and CTE. but to it is not clear that without iid how do you calculate the feature values. if SHAP + i.i.d sampling is not the baseline, the please clearly explain how you calculated the baseline feature values? what is the baseline for that you compared the iid and cte sampling? (Table 1)

---

> ### Author Response · Authors · 2024-11-15
> **Response by Authors [1st out of 2]**
>
> **We gratefully thank the reviewer for a thorough and very insightful review of our work. Below, we provide point-by-point responses to each of the comments.**
>
> > This approach significantly reduces the computational complexity of SHAP and SAGE.
>
> Thank you for highlighting the main strength of our approach. Beyond **reducing the computational cost** of SHAP and SAGE, our proposed CTE framework significantly **improves the stability** of estimating SHAP and SAGE (see **Table 1 & Figure 6**). Also, the reviewer may have overlooked the fact that the approach is more general, i.e. it can be applied to other post-hoc explanation methods. **We concretely study 2 more methods** in the paper, namely **Feature Effects** (**Figure 7**) and **Expected Gradients** for image (**Figure 5**) or, in general, unstructured data (**Figure 13** in Appendix F).
>
> > **W1:** line 53 - "We introduce a new paradigm for estimating post-hoc explanations based on a marginal distribution" ---> The main contribution of paper is not clearly described. From the introduction, reader might get the idea that you are proposing a new explanation method that employs KT, but as we go further into the paper, the tone of authors changes and they focus on the effectiveness their approach for Kernel SHAP and SAGE.
>
> **A1:** Thank you for pointing this out. We rephrased the original sentence in **lines 53 & 74** from "We introduce a new paradigm for estimating post-hoc explanations based on a marginal distribution compressed more effectively than with i.i.d. sampling." to "We introduce a new paradigm of sample-efficient explainability where post-hoc explanations, like feature attributions and effects, are estimated based on a marginal distribution compressed more efficiently than with i.i.d. sampling."
>
> > **W2:** the idea of this paper needs to be more clearly stated. From my understanding the paper attempts to improve the computation time of SHAP and SAGE using Kernel thinning. The Kernel Thinning is proposed to better select the samples for the feature attribution explanation. The idea seems interesting and this can help with the Shapely and Sage method, but this approach is only specific to these explanation method that are based on the selection of subsets of data.
>
> **A2:** Yes, our approach is specific to **a broad class of explanation methods** that are based on the selection of subsets of data. Examples of such explanations improved by our approach are SHAP and Expected Gradients (feature attributions), SAGE (feature importance), and partial dependence (Feature Effects). We aim to clarify this fact across the following:
> - Section 1, **lines 34–41** and **contribution (1)**: "We bring to attention and measure the approximation error introduced by using i.i.d. sampling of background and foreground data in various explanation methods."
> - **Section 2.1** titled "Sampling from the dataset is prevalent in explanation estimation",
> - as well as **Appendix A** "Motivation: standard i.i.d. sampling in explanation estimation" where examples of other explanation methods are given.
>
> > **W3:** The experimental evaluation is heavily focused on MAE with the explanation as "...". I wonder if this approach is the main criteria that paper investigated, what is the main *baseline* that they compare with to obtain the explanation error. If the explanation error is calculated with the SHAP with the iid sampling, so what does Table 1 mean. because in this table the paper explains that from the iid sampling the MAE of explain improved to something that is achieved with CTE.
>
> **A3:** Thank you for this question; let us recall the details of our experimental setup given at the beginning of **Section 4**. The *baseline* that the 'sampled' explanation is compared with to obtain the estimation error is the **'ground truth' explanation** estimated using a full validation dataset (see **lines 269 & 290–293** in the paper). **Table 1** shows the difference in estimation error between i.i.d. sampling and CTE, where **for each of them**, the error is calculated **separately** as a difference between the explanations estimated on sampled (i.i.d. or CTE) and complete data (see the formulas in **lines 296–299**). Please note that **Section 4.1** also reports values of the **Top-k metric**, measured with respect to the 'ground truth', similarly to MAE. **Top-k values** show that compression is better than sampling in preserving the ranking of feature attributions and importance (see **Figure 2B**).

---

> ### Author Response · Authors · 2024-11-15
> **Response by Authors [2nd out of 2]**
>
> > **W4:** Also the paper only explored the Gaussian Kernel. Why that choice? why not exploring other kernels? What is the theoretical aspect of Gaussian that advantages your approach?
>
> **A4:** This is a great question, for which we have now added an answer in the paper **lines lines 522–526**: "We used the Gaussian kernel because it is the standard in the field of distribution compression (Dwivedi & Mackey, 2022; Shetty et al., 2022; Domingo-Enrich et al., 2023), especially in experimental analysis, and is generally adopted within machine learning applications. Although our empirical validation shows that the Gaussian kernel works well for over 50 datasets, exploring other kernels for which theoretical thinning error bounds exist, like Matern or B-spline (Dwivedi & Mackey, 2021), is a viable future work direction."
>
> > **W5:** also, I wanted to see some examples of how the trained explainer using your method performs for image and examples of IMDB dataset. The paper heavily focuses on the MAE without providing any example to show the qualitative comparison of generated explanation
>
> **A5:** Please note that we show an example for the IMDB dataset in **Appendix F, Figure 14D**. While we are fine with showing concrete visual examples – which is why we already provided the one exactly on the dataset you requested – we would also like to politely point out that, generally, the **qualitative comparison of generated explanations is prone to human confirmation bias** and could be studied by experts in HCI. Instead, we thoroughly evaluate estimation accuracy with the MAE and **Top-k metrics**. We provide more examples, also for images, in code notebooks that can be found in the **Supplementary Material** (see the **'examples' directory**). In summary, we believe the requested aspect is already addressed. Kindly review it again.
>
> > **W6:** The main purpose of KT, is reducing the computational complexity and the quality of generated explanation, but since this method is focused on the data attribution concept, I was expecting see a comparison between the KT and Influence Function in extracted explanations.
>
> **A6:** While we (truly) think that this is indeed an interesting direction, **we think that it does not invalidate our method and positive results**. To the best of our knowledge, there currently is no overlap between the two concepts (distribution compression and data attribution) studied in the literature. The main purpose of KT is **an efficient approximation of data distribution**, while Influence Functions aim to **attribute the importance of data to the model’s error**. While one could construct an alternative approach based on this (vaguely) formulated direction, one would have to concretely and carefully analyze this first. We did such an analysis for our approach instead.
>
> > **Q1:** line 184 - definition 4 - it is not clear what is the output f this function and how do you compare it with for calculation of MAE. how the explanation error is defined and how it is calculated? as far as I understand the mean absolute error is measured based on the difference between the assigned values to features of explanation from the SHAP with iid and CTE. but to it is not clear that without iid how do you calculate the feature values. if SHAP + i.i.d sampling is not the *baseline*, the please clearly explain how you calculated the *baseline* feature values? what is the *baseline* for that you compared the iid and cte sampling? (Table 1)
>
> **A7:** In **Definition 4**, the output of the function $f$ is the model’s prediction with **marginalized features**, which is defined in **Definition 1** (feature marginalization). The *baseline* that the 'sampled' explanation is compared with to obtain MAE is the **'ground truth' explanation** estimated using a full validation dataset (see **lines 269 & 290–293** in the paper). **Table 1** shows the difference in MAE between i.i.d. sampling and CTE, where **for each of them**, the error is calculated **separately** as a difference between the explanations estimated on sampled (i.i.d. or CTE) and complete (not sampled) data (see the formulas in **lines 296–299**). The general explanation error is denoted in **Equations 1 & 2** with a norm, which is then explicitly written as MAE for each explanation method: in **lines 297–298** for SHAP & Expected Gradients, and in **lines 298–299** for SAGE.
>
> **Please let us know if our above responses clarify the raised concerns.**

---

> ### Author Response · Authors · 2024-11-20
> **Thanks again for your feedback**
>
> Dear Reviewer fa33,
>
> Thanks again for your feedback. We have addressed all your points and hope our response resolves any concerns. **We would greatly appreciate an updated score and are happy to discuss any remaining concerns.** Your input is invaluable, and we genuinely appreciate your time and effort.
>
> Best, Authors

---

> > ### Comment · Reviewer_fa33 · 2024-11-29
> >
> > I want to thank the authors to take time and revised the paper to include the qualitative results in the paper. I have raised my score.

---

### Author Response · Authors · 2024-11-15
**Response by Authors**

**We gratefully thank the anonymous reviewers for their time and effort in reviewing our paper.** We are excited about the appreciation of our work and the valuable feedback to improve it. In separate comments, we provide point-by-point responses to each of the reviews.

Here, we additionally list all the changes made in the edited version of the manuscript, **denoted in green text**.

As suggested by **Reviewer fa33**:

1. We rephrased the original sentence in **lines 53 & 74** from "We introduce a new paradigm for estimating post-hoc explanations based on a marginal distribution compressed more effectively than with i.i.d. sampling." to "We introduce a new paradigm of sample-efficient explainability where post-hoc explanations, like feature attributions and effects, are estimated based on a marginal distribution compressed more efficiently than with i.i.d. sampling."
2. We added a comment on alternative kernel functions in **lines 522–526**: "We used the Gaussian kernel because it is the standard in the field of distribution compression (Dwivedi & Mackey, 2022; Shetty et al., 2022; Domingo-Enrich et al., 2023), especially in experimental analysis, and is generally adopted within machine learning applications. Although our empirical validation shows that the Gaussian kernel works well for over 50 datasets, exploring other kernels for which theoretical thinning error bounds exist, like Matern or B-spline (Dwivedi & Mackey, 2021), is a viable future work direction."

As suggested by **Reviewer dB4i**:

3. We labeled sub-plots as “A), B), C), …” and extended the captions in **Figures 2, 4, 6 & 8** to improve the presentation.

---

> ### Author Response · Authors · 2024-11-27
> **Additional changes made in the edited version of the manuscript**
>
> As suggested by **Reviewer fa33**:
>
> 4. We added exemplary visual comparisons between ground truth explanations and those estimated on an i.i.d. and compressed sample in **Appendix I**.
> 5. We extended the discussion of the results for CIFAR-10 and MNIST in the paper **lines 397–401**.
> 6. We added a comment on influence functions in the paper **lines 529–532**: “One could also investigate how influence functions (Koh & Liang, 2017), which aim to attribute the importance of data to the model’s prediction, can guide sampling for efficient explanation estimation.”

---

### Meta-Review · Area_Chair_eoP6 · 2024-12-20

**Metareview:**

The paper introduces a compress-and-explain scheme to speedup attribution calculation. The reviewers are largely in agremeent that the proposed method is worthy of publication.

**Additional Comments On Reviewer Discussion:**

No discussions needed because the reviewers are in agreement.

---

### Decision · Program_Chairs · 2025-01-22

Accept (Spotlight)